# Preserved extrastriate visual network in a monkey with substantial, naturally occurring damage to primary visual cortex

Holly Bridge[1,2]*, Andrew H Bell[1,3,4], Matthew Ainsworth[3,4], Jerome Sallet[1,3], Elsie Premereur[5], Bashir Ahmed[6], Anna S Mitchell[3], Urs Schüffelgen[1,3], Mark Buckley[3], Benjamin C Tendler[1,2], Karla L Miller[1,2], Rogier B Mars[1,2,7], Andrew J Parker[6], Kristine Krug[6]*

[1]Wellcome Centre for Integrative Neuroimaging, FMRIB, Oxford University, Oxford, United Kingdom; [2]Nuffield Department of Clinical Neurosciences, Oxford University, Oxford, United Kingdom; [3]Department of Experimental Psychology, Oxford University, Oxford, United Kingdom; [4]MRC Cognition and Brain Sciences Unit, Cambridge, United Kingdom; [5]Laboratory for Neuro- and Psychophysiology, KU Leuven, Leuven, Belgium; [6]Department of Physiology, Anatomy and Genetics, Oxford University, Oxford, United Kingdom; [7]Donders Institute for Brain, Cognition and Behaviour, Radboud University Nijmegen, Nijmegen, Netherlands

*For correspondence:
holly.bridge@ndcn.ox.ac.uk (HB);
kristine.krug@dpag.ox.ac.uk (KK)

**Abstract** Lesions of primary visual cortex (V1) lead to loss of conscious visual perception with significant impact on human patients. Understanding the neural consequences of such damage may aid the development of rehabilitation methods. In this rare case of a Rhesus macaque (monkey S), likely born without V1, the animal's in-group behaviour was unremarkable, but visual task training was impaired. With multi-modal magnetic resonance imaging, visual structures outside of the lesion appeared normal. Visual stimulation under anaesthesia with checkerboards activated lateral geniculate nucleus of monkey S, while full-field moving dots activated pulvinar. Visual cortical activation was sparse but included face patches. Consistently across lesion and control monkeys, functional connectivity analysis revealed an intact network of bilateral dorsal visual areas temporally correlated with V5/MT activation, even without V1. Despite robust subcortical responses to visual stimulation, we found little evidence for strengthened subcortical input to V5/MT supporting residual visual function or blindsight-like phenomena.
DOI: https://doi.org/10.7554/eLife.42325.001

## Introduction

Primary visual cortex (V1) of primates is the major gateway for feedforward input of visual information from the retina via the lateral geniculate nucleus (LGN) into a network of over 30 extrastriate visual areas (*Felleman and Van Essen, 1991*; *Markov et al., 2014*; *Schmidt et al., 2018*). V1 contains a complete, high resolution retinotopic map and contributes to cortical processing by computing local spatio-temporal correlations of the input, which is evident in its neural representations of local visual features (orientation, spatial frequency, temporal frequency, direction, colour, binocular disparity) (e.g. *Hubel and Wiesel, 1959*; *Movshon et al., 1978*; *Parker et al., 2016*). The direct contribution of V1 signals to conscious sight is a subject of ongoing scientific debate (e.g. *Stoerig, 2006*; *Ffytche and Zeki, 2011*) and is thought to involve feedback as well as feedforward input to V1 (*Ress and Heeger, 2003*). At the centre of this debate have been patients with V1 lesions exhibiting

**eLife digest** Vision depends on the brain as well as on the eyes. Almost all information from the eyes travels to a brain region called the primary visual cortex. This large expanse of tissue at the back of the brain contains a detailed map of the visual world. Adults who suffer damage to part of the primary visual cortex become blind in the corresponding area of visual space, a phenomenon known as cortical blindness. Yet, adults with cortical blindness can also experience 'blindsight': they can still point correctly to bright, moving images, even though they claim they cannot see them.

One of the roles of primary visual cortex is to act as a gateway to other, 'higher' visual areas of the brain. These regions process the input they receive from the primary visual cortex to generate a rich and coherent visual representation. But how do adults with blindsight, in whom the major gateway from the eyes to higher visual areas has been damaged, still manage to respond to visual stimuli?

By chance, Bridge et al. discovered a monkey whose unusual brain anatomy provides clues as to why this is possible. The monkey behaved much like its peers, leaping between the perches of its enclosure with ease. But when Bridge et al. tried to train the animal on a visual task, it proved unable to learn like the other monkeys. A brain scan revealed that it had almost no primary visual cortex, probably because of an abnormality that arose early in development.

Further studies of the monkey's brain showed that the other structures involved in visual processing were in their usual places and were of normal size. Connections from the eyes to higher visual areas that bypass the primary visual cortex were normal, but were no stronger than in other monkeys. In fact, areas beyond the primary visual cortex showed no fundamental changes in how they processed visual information.

The brain of this monkey had thus adapted to early loss of the major gateway from the eyes to higher visual cortex. Visual areas of the brain beyond the primary visual cortex continued to work as normal, helping to minimize vision loss. Because the visual brain differs little between primates, this discovery could also benefit patients with blindsight. It suggests that targeting higher visual areas could further improve vision in patients with damage to the primary visual cortex.

DOI: https://doi.org/10.7554/eLife.42325.002

residual vision - often without visual awareness - a condition termed Blindsight (*Riddoch, 1917*; *Weiskrantz et al., 1974*; *Cowey, 2010*).

Previous data on bilaterally cortically blind macaque monkeys suggest that there is a dissociation between use of vision for guiding movement and for awareness and recognition (*Leopold, 2012*). Specifically, monkey Helen, who had V1 removed bilaterally, was able to navigate around the world but unable to recognise faces or food (*Humphrey, 1974*). The lesions to Helen were made in adulthood, which may have affected the amount of residual vision. Macaque monkeys who received a unilateral V1 lesion at two months of age exhibited more residual vision as adults than monkeys who received their lesions in adulthood (*Moore et al., 1996*).

Cortical blindness due to bilateral damage to the primary visual cortex of humans is fortunately rare. There are a few cases of damage acquired in adulthood, some of whom have been extensively studied (*de Gelder et al., 2008*; *Hervais-Adelman et al., 2015*; *Arcaro et al., 2018*), but also a number of children who acquired lesions congenitally or through perinatal stroke (*Mundinano et al., 2017*). The presence of residual visual function suggests that functional visual networks can develop or be sustained in the absence of the main visual input to cortex.

In monkeys, in the absence of primary visual cortex, a number of potential pathways have been proposed to convey visual information from the eyes to extrastriate visual cortex. Direct input to extrastriate visual cortex bypassing V1 may arise from the lateral geniculate nucleus (LGN) or the pulvinar nucleus directly to extrastriate visual areas V2, V3, V4, V5/MT and inferotemporal cortex (*Wong-Riley, 1976*; *Sincich et al., 2004*; *Kaas and Lyon, 2007*; *Warner et al., 2010*; *Gattass et al., 2014*). In humans, both pulvinar and LGN inputs to V5/MT have been implicated (*de Gelder et al., 2008*; *Ajina et al., 2015b*). In addition to its direct retinal input, the LGN also receives projections from the superior colliculus (SC) (*Harting et al., 1991*; *Stepniewska et al., 1999*). The pulvinar nucleus receives input from the SC, although likely not into the appropriate subdivision for

projectiions to V5/MT, and pulvinar might receive surviving early projections directly from the retina (*Stepniewska et al., 1999*; *Warner et al., 2010*). It has been suggested that early-life V1 lesions in particular might lead to less pruning of the retina-pulvinar-V5/MT pathway and that this might contribute to 'blindsight' (*Warner et al., 2015*).

Macaque monkey S in the current study showed bilaterally enlarged lateral ventricles that appeared to have expanded into space usually occupied by V1. This gross pathology of the cerebral cortex was most likely to have been acquired developmentally or perinatally. Previous data have implicated roles in residual vision for the LGN, pulvinar, SC and the extrastriate visual cortical networks including motion area V5/MT. Our aim here was to use magnetic resonance imaging (MRI) to determine the integrity of the structural and functional networks underlying residual visual function in this monkey with a long-standing anatomical lesion of primary visual cortex.

## Results

### Case history of monkey S

A female Rhesus macaque (monkey S; 7 years; 6.25 kg) initially showed comparable behaviour to cage mates, and successfully followed the same familiarisation and training pattern for coming out from the home enclosure into a transport box or primate chair (*Mason et al., 2019*). Training in both cases was completed in a comparable time frame to other monkeys trained at the same time. Subsequently, task training took place five days per week over 18 months – this involved visual tasks but was intended for a decision-making study, rather than for measurement of visual performance. Training began with a simple task in a transport box, 'anytouch', where monkey S was rewarded for touching the screen anywhere during presentation of an array of colourful typographical characters. Monkey S managed to complete 10 trials in 3 min within one week of training and thus progressed to the next training stage. Task 'oneplace' involved targeted touching of a single coloured typographical character at a random location. Monkey S reached 40–50 trials in increments of 10 trials, but continued with the lowest complexity of visual presentations over six weeks of training. The 40–50 trials were completed in a time window ranging between 30–91 min per session. This was a particularly poor performance compared with other monkeys progressing through this auto-shaping programme. Monkey S was unable to progress further from the initial stages of 'oneplace'.

At this phase of the training, monkey S was transferred to a primate chair (seated, but without head clamp or eye movement control) and to a different reward schedule (juice instead of pellets), and 'oneplace' was attempted again. Monkey S still worked much more slowly than other monkeys, attaining 40 rewarded trials in 33 min over many weeks of training sessions, whereas typical performance is expected to be closer to 100 rewarded trials in 18–25 min. Monkey S would typically no longer respond to the task when the number of trials or the complexity of the task were increased.

We tried using different effectors (e.g. a response lever and a metal knob) in an attempt to improve task learning and performance. Initially, monkey S was trained to hold the lever and then release it for reward. Although she did well during this initial behavioural training, monkey S was unable to learn to release the lever in response to visual stimulation. When monkey S was required to do a simple spatial search using the lever in response to a change in target colour on the screen, she typically only completed up to 10–20 trials towards the end of six months of training and only when the trainer remained present in the room. During this time, rudimentary checks determined that, while seated in the primate chair, monkey S could orient towards rewards and objects when held up in front of her and when moved around in her visual field.

Finally, monkey S was returned to the transport box touch screen training with 'oneplace'. She completed up to 40 trials for banana pellets after 6 weeks of training, but still required 41–103 mins to complete a session, making between 23–51 errors. An error was recorded when monkey S touched the touch screen anywhere apart from the alphanumeric character.

In spite of the inability of S to perform these tested psychophysical tasks, behavioural assessment in the home enclosure by a clinical neurologist and two neuroscientists showed no qualitatively different patterns of locomotion and visual orienting towards other monkeys and people in the room. There was some indication that patterns of eye fixation might be atypical, but no quantitative assessment was undertaken. However, when monkey S was offered treats, she would start repeatedly

running past the treat and pick it up with a sideways reach while running - rather than coming forwards, fixating the treat and reaching for it in a more controlled way.

As monkey S so obviously differed from the animals co-housed with her, we performed a brain scan. Structural MRI revealed an almost complete loss of primary visual cortex (V1) (*Figure 1*, top), consistent with presumptive bilateral visual field loss (see *Figure 1—figure supplement 1*). Clinical assessment of the MRI scans and records suggested no injury or history that could explain this lesion and that this was probably a congenital or perinatal condition.

## Subcortical visual areas are structurally normal in monkey S

The LGN, the pulvinar, the SC and V5/MT are four structures that have been commonly thought to support residual vision in blindsight. Therefore, our MRI analysis of the visual brain focussed more specifically on those four brain structures.

In humans with lesions of V1, which result in hemianopia, the LGN is often reduced in size, due to retrograde degeneration (*Miki et al., 2005*; *Bridge et al., 2011*). A similar result has also been found in the adult marmoset (*Atapour et al., 2017*). When we investigated the structure of the LGN in monkey S and four female control Rhesus macaques of a similar age with intact visual systems, there was no obvious reduction in size of the structure from inspection of the images (*Figure 1*). The volume of the LGN in the control monkeys was measured at 50.0 mm$^3$ ± 4.7 mm$^3$ and 51.5 mm$^3$ ± 5.1 mm$^3$ (mean ± SD) for the left and right hemispheres. In monkey S, the comparable values were 47.0 mm$^3$ and 48.5 mm$^3$ respectively. Thus, the structure seems not to have been affected by retrograde degeneration. Similarly, the superior colliculus (SC) was also comparable in size in monkey S (left = 30.0 mm$^3$; right = 31.6 mm$^3$) and control monkeys (left = 31.0 mm$^3$ ± 1.9 mm$^3$ and right = 30.5 mm$^3$ ± 3.3 mm$^3$) (mean ± SD). Finally, the pulvinar was also of a similar size in monkey S (left = 40.2 mm$^3$; right = 41.9 mm$^3$) and control monkeys (left = 37.7 mm$^3$ ± 4.4 mm$^3$ and right = 35.9 mm$^3$±3.2 mm$^3$) (mean ± SD).

## Cortical structure: V1 appears much thinner in monkey S but area V5/MT shows similar pattern of dense myelination to control monkeys

To investigate the cortical changes in more detail, we acquired *post mortem* T2*-weighted images in monkey S and for the right hemisphere of a control monkey (M131) (*Figure 2*). The stripe of Gennari is visible in both monkey S and the control monkey. However, the stripe is closer to the pial surface in monkey S, rather than in the middle of the cortical ribbon, as seen in the control monkey. This could suggest that the superficial layers are particularly reduced as a consequence of the extensive lesion. Measurement of cortical thickness in V1 in both monkeys indicated that monkey S has substantially thinner cortex around the lesion. By contrast, extrastriate visual area V5/MT appeared to have a thickness of about 2 mm, which is comparable to the control monkey.

T1w/T2w structural MRI images provide a qualitative signal indicative of myelination within the cortical ribbon in vivo (*Glasser and Van Essen, 2011*; *Large et al., 2016*). To investigate whether there were structural consequences of the loss of major feedforward input to extrastriate visual cortex, we took such scans from monkey S and compared the results with those previously obtained from the same four female control Rhesus monkeys as in the previous section (*Figure 3*). These myelin-weighted images show a distinct band indicative of dense myelination in the lower layers of extrastriate cortical area V5/MT on the posterior bank of the dorsal superior temporal sulcus (STS), but not in adjacent cortical areas. This band is similar in location and extent to the four control monkeys. We measured the mean surface area of this region of distinct myelination (top third of image intensity values) in dorsal STS, likely to comprise V5/MT and MST, as 86.9 mm$^2$ (left hemisphere) and 78.8 mm$^2$. This is in the range of published data of 82.4 mm$^2$ (SD = ±8.9; n = 10 hemispheres) obtained by the same method (*Large et al., 2016*).

## LGN shows significant activation to visual stimulation in monkey S

We conducted eight scanning sessions in seven Rhesus macaque monkeys. First, we analysed the BOLD responses to a contrast-reversing, full-field flickering checkerboard. We then restricted all further analyses to those monkeys that had significant BOLD activation in the LGN (z > 2.3) to this stimulus (see Materials and methods and *Figure 4*; *Figure 4—figure supplement 1*). One scanning

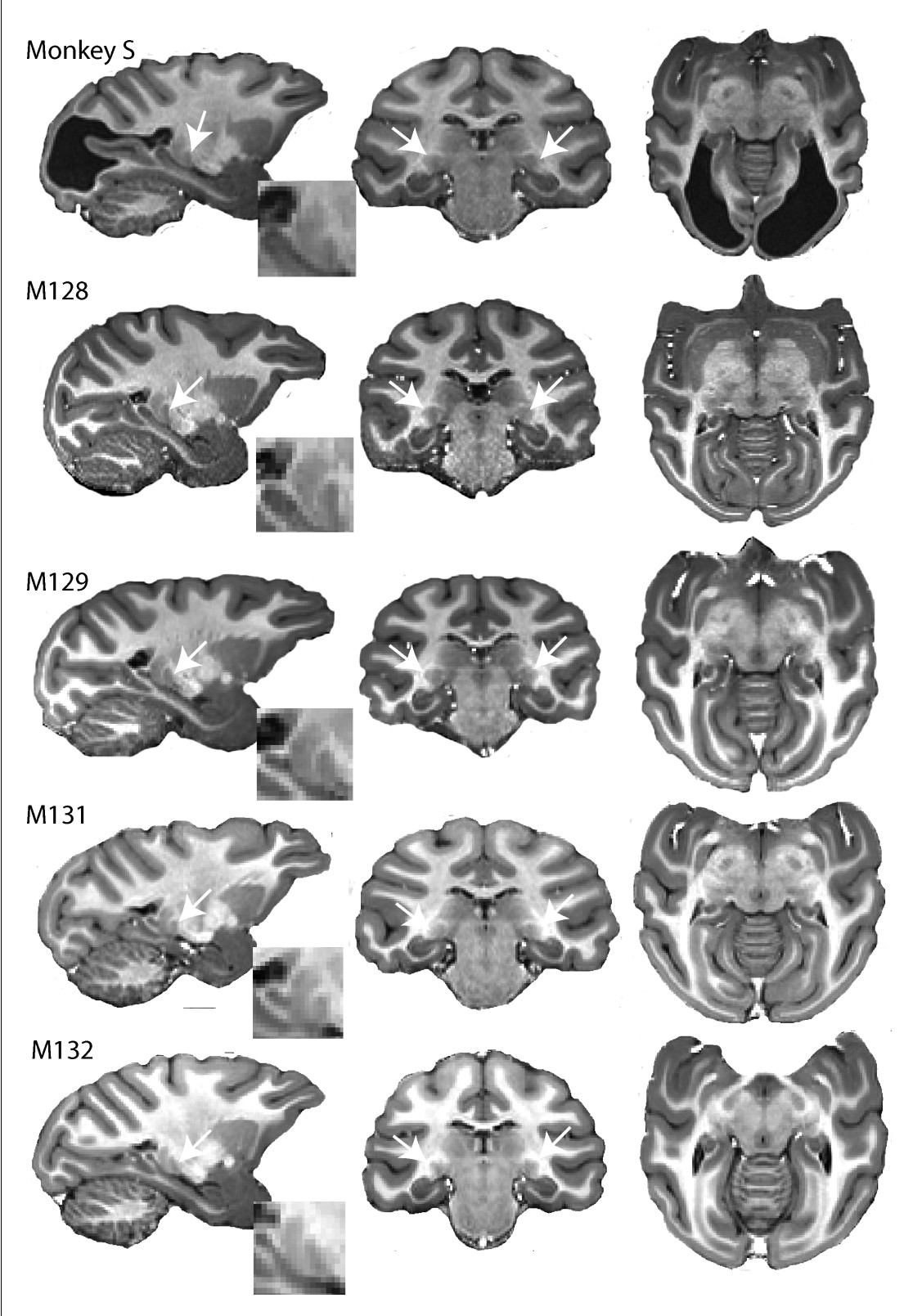

**Figure 1.** High resolution structural images of five adult female macaques of similar age and weight. The large bilateral lesions of monkey S to the occipital cortex are clearly visible. LGN (white arrow) and pulvinar (posterior and dorsal to LGN) are shown in parasagittal section (with high magnification inset) as well as in the coronal and horizontal sections.

DOI: https://doi.org/10.7554/eLife.42325.003

*Figure 1 continued on next page*

*Figure 1 continued*

The following figure supplement is available for figure 1:

**Figure supplement 1.** Visual map estimate of monkey S.
DOI: https://doi.org/10.7554/eLife.42325.004

session from each of four monkeys fulfilled this criterion, including the first of two separate scanning sessions with monkey S.

*Figure 4* shows the BOLD activation in response to the flickering checkerboard compared to a mid-grey screen. The strong activation evident in the LGN of monkey S, is consistent with the intact structure shown in *Figure 1*, and suggests that information from the retina can reach the brain. The

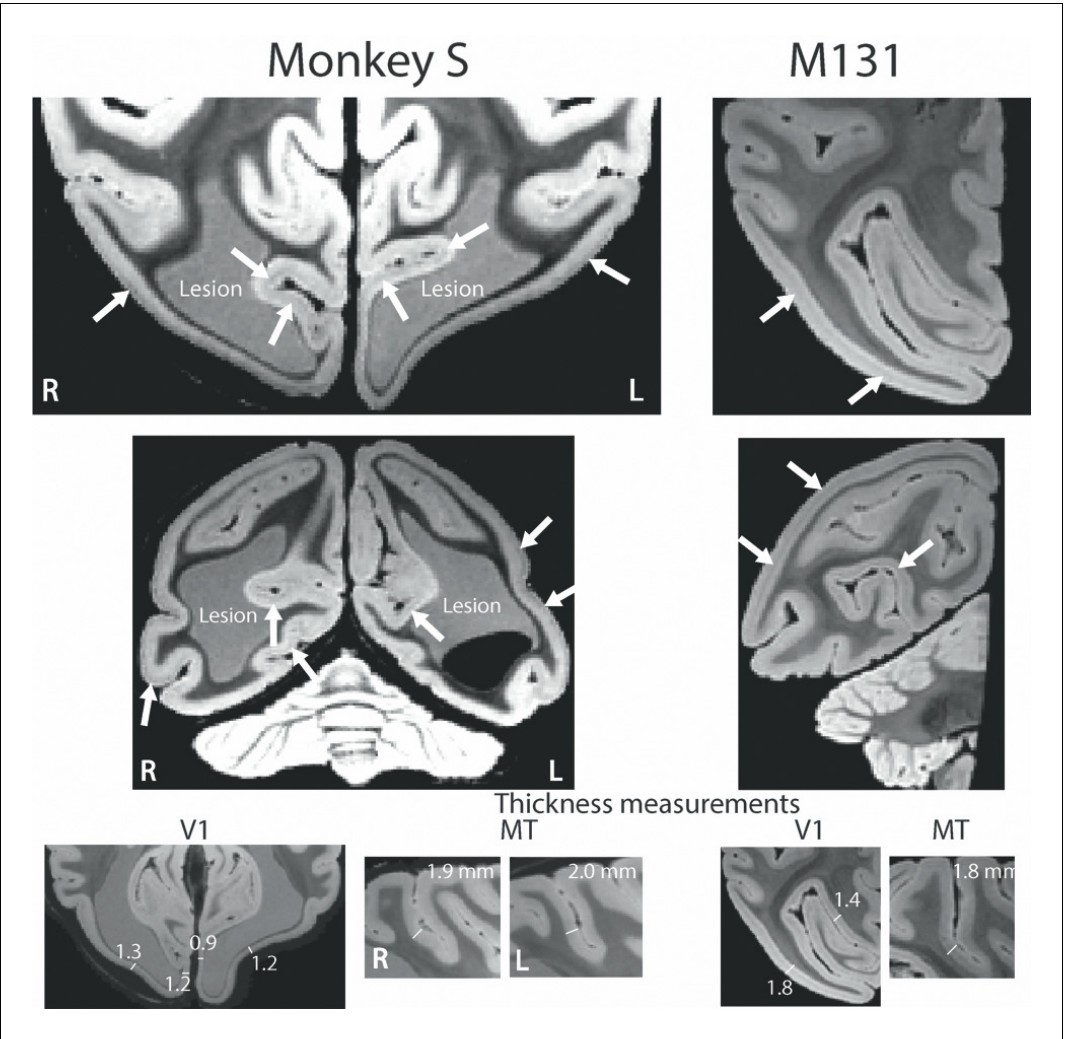

**Figure 2.** *Post mortem* high resolution scan of V1. A *post mortem* T2*-weighted structural scan at high field (7T) revealed the stripe of Gennari in V1 (white arrows for example sites) of both monkey S and a control monkey (M131), although the position relative to the cortical surface within the cortical ribbon appeared to differ. The V1 grey matter in monkey S was reduced in thickness to 0.9–1.3 mm around the enlarged ventricles (1.4–1.8 mm in the control). The appearance of increased thickness more laterally in the sulcus is due to the angle of the slice through cortex at this point. By contrast, the grey matter at the location for V5/MT appeared intact with a typical thickness of 1.9–2.0 mm. Note that the lesion is partially light grey in appearance as well partially black. As the brain was still inside the skull during the scan, we speculate that this might be a result of slow incursion of 'fluorinert', in which the skull was immersed for this scan.
DOI: https://doi.org/10.7554/eLife.42325.005

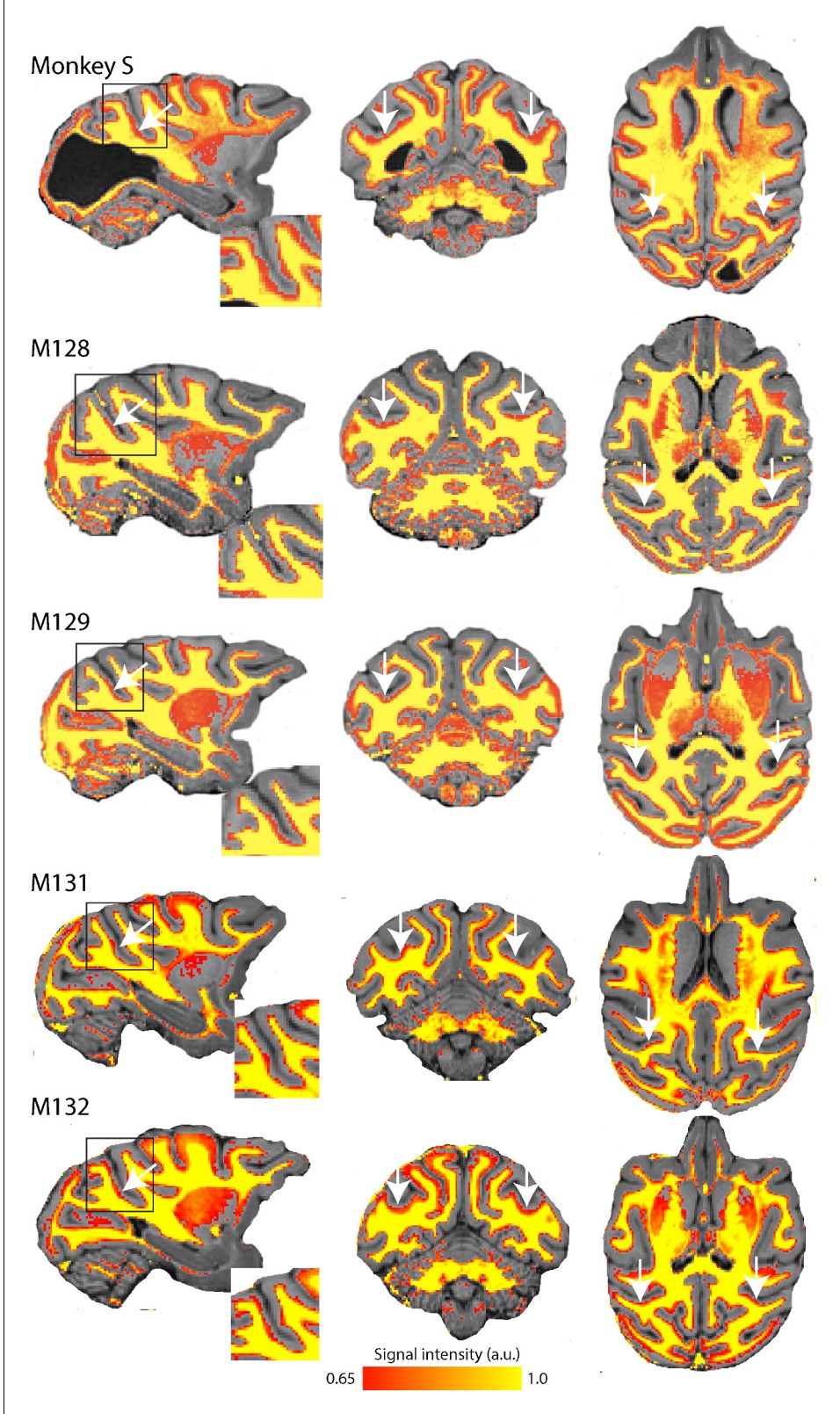

**Figure 3.** Myelin-weighted T1w/T2w images obtained in vivo show the expected pattern of dense myelination. Yellow represents the highest intensity signal in the images, corresponding to the dense white matter, while red is less dense white matter, corresponding either to myelin within the cortical ribbon, as in the case of area V5/MT, or subcortical grey matter. The white arrows indicate the location of V5/MT in each view, and the red voxels within the cortical ribbon indicate the increased myelination expected in controls and monkey S.

*Figure 3 continued on next page*

*Figure 3 continued*

DOI: https://doi.org/10.7554/eLife.42325.006

control monkeys included in these analyses showed similar activation in the LGN in response to this stimulus. In contrast to the strong activation in the LGN, there was less activation evident in visual cortical regions, especially in the primary visual cortex of monkey S (see *Figure 4—figure supplement 2* for a complete series of sections). Even in the control monkeys there was considerable variability in the location and extent of cortical BOLD activation. The checkerboard stimulus is expected to activate V1, but even in some control animals where V1 is intact, the activation was not extensive. Some of this variability may be due to effects of anaesthesia on visual responses (see *Figure 4—figure supplement 1*; see Discussion).

While the checkerboard stimulus is designed to activate early visual areas, a moving dot stimulus may evoke greater activation in some extrastriate areas, particularly where there is damage to early areas. The responses to moving dots compared to stationary dots are shown in *Figure 5*. The level of activation to this type of stimulation was lower in all animals, including monkey S. Control monkey M901 showed very little activation to the moving dots in any brain region, in contrast to the strong activation evident in the LGN to flickering checkerboards. Monkey S showed some LGN activation on the left side of the brain and in the pulvinar situated posterior to the LGN. Cortical activation was sparsely distributed around the brain, with some activation within the dorsal part of the Superior Temporal Sulcus (STS) near area V5/MT in monkey S (see *Figure 5—figure supplement 1*) and in control monkeys M142 and M902.

The whole brain analyses suggested that activation was greatest to the flickering checkerboard, particularly in the LGN. To quantify activation and compare responses in monkey S with the control monkeys, the % BOLD signal was extracted from the LGN, pulvinar, V1 and area V5/MT. Masks were defined anatomically on the brains of each individual monkey in structural space using a standard atlas as a guide (National Institute of Mental Health Macaque Template - NMT) (*Seidlitz et al., 2018*). The activation levels in *Figure 6* support the observation that the activity in LGN is greatest to the checkerboards across all animals, including monkey S. For control monkeys, V1 activation tended to be lower than for the LGN, with a median ratio of V1:LGN of 0.83 (range 0.2–0.92), but this was 0 for monkey S as she showed no V1 activation. Furthermore, monkey S did not show significant activation in anatomically defined area V5/MT to either stimulus but did in the pulvinar, though only in response to the moving dots. The timeseries from the LGN (checkerboard) and the pulvinar (moving dots) activation for monkey S illustrate the variability of the BOLD signal in these regions (see *Figure 6—figure supplement 1*). The only other monkey with activation in the pulvinar to the moving dots (M902) showed similar activation to the checkerboard stimulus, a pattern not seen in monkey S.

The pathways most often proposed to underlie blindsight in human patients with hemianopia include the visual motion complex hMT+ (thought to comprise visual areas V5/MT and MST) and this region often shows significant activation in response to moving stimuli, similar to the stimulus used here (*Ajina et al., 2015a*; *Ajina and Bridge, 2016*), and to contrast defined stimuli (*Ajina et al., 2015c*). Thus, it was surprising that anatomically defined V5/MT in monkey S did not show significant activation to either type of stimulation. To investigate the activation patterns in monkey S in more detail, *Figure 7* shows a series of slices, 3 mm apart, through dorsal aspects of the superior temporal sulcus (STS) including visual areas of hMT+ in the human visual system (see also *Figure 5—figure supplement 1* for a 1 mm series of sections through the whole brain of monkey S). While there was no spatially extensive region of activation, there were a number of regions within the sulcus showing BOLD activation (z > 2.3) to the moving dot stimulus, including in area MST on the anterior bank and area FST at the bottom of the sulcus, but mainly outside V5/MT.

## Area V5/MT shows a normal pattern of functional connectivity within the cortex

Since the monkeys were all anaesthetised, it was difficult to determine whether the scarcity of cortical activity in monkey S was due to effects of the anaesthetic or reflected a real difference in signal processing. For instance, it is conceivable that the heavily enlarged, fluid-filled ventricles within the

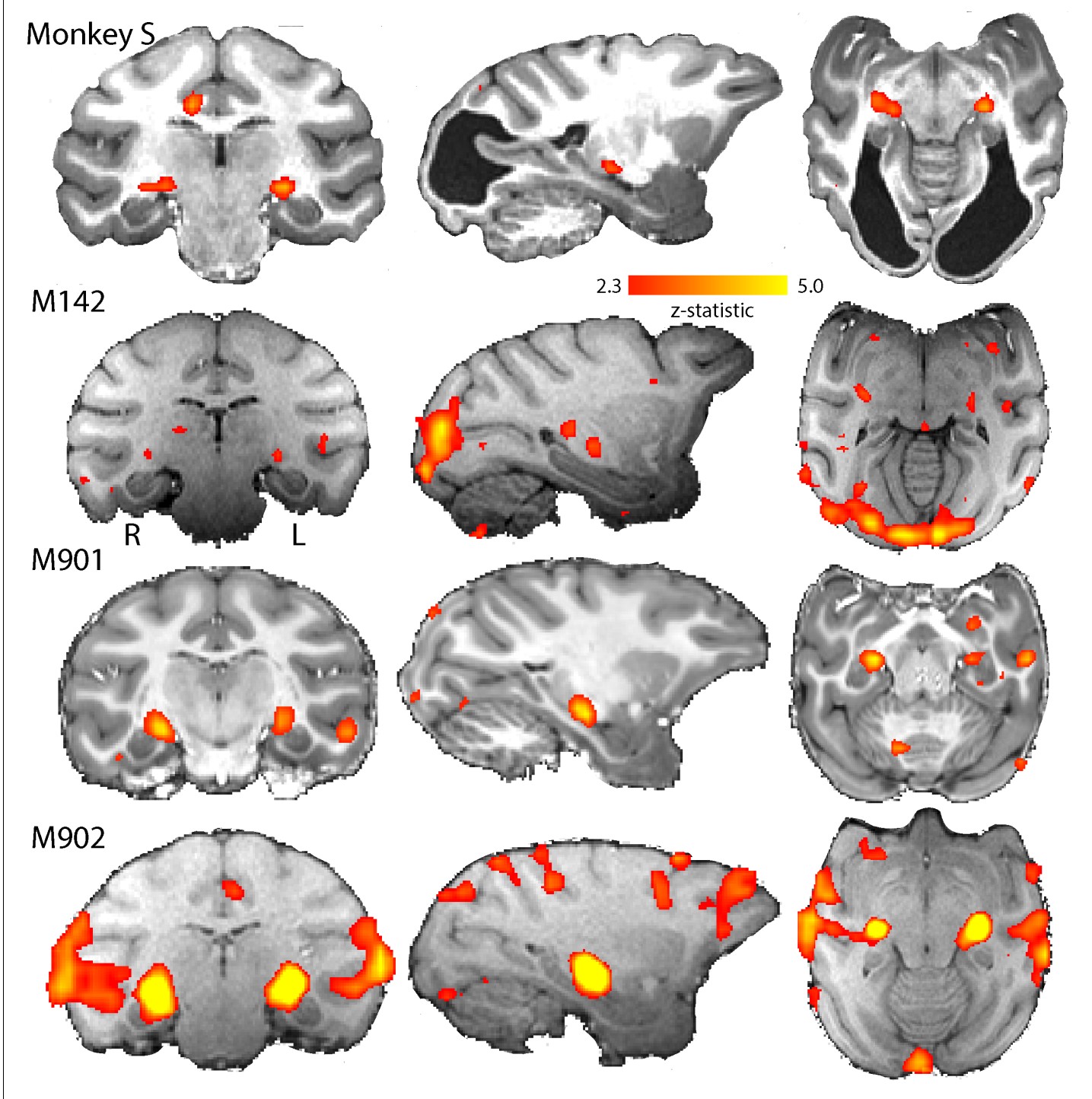

**Figure 4.** BOLD Activation by high contrast stimuli. LGN was significantly activated in all monkeys by flickering checkerboards (z > 2.3). Cortical activation is less consistent across the control monkeys and not really evident in monkey S. The parasagittal sections show that the pulvinar was not consistently activated by this stimulus.

DOI: https://doi.org/10.7554/eLife.42325.007

The following figure supplements are available for figure 4:

**Figure supplement 1.** Anaesthesia, weight, and significant visual responses.

DOI: https://doi.org/10.7554/eLife.42325.008

**Figure supplement 2.** BOLD activation by high contrast stimuli for monkey S.

*Figure 4 continued on next page*

*Figure 4 continued*

DOI: https://doi.org/10.7554/eLife.42325.009

visual cortex that formed the lesion could have differentially affected the pattern of action of anaesthesia in monkey S. *Figure 6* showed no significant V5/MT activation in monkey S to either checkerboard or moving dots stimuli, so to understand the nature of the response in area V5/MT, we performed a connectivity analysis. Rather than using the amplitude of BOLD activation, this analysis uses the fluctuations in the signal over time to determine areas that are likely to have common inputs or to be connected. This method was used previously to investigate the functional connectivity patterns of temporal and frontal regions in macaque monkeys (*Vincent et al., 2007*; *Sallet et al., 2013*; *Hutchison et al., 2012*; *Mars et al., 2013*).

*Figure 8* shows the regions of the cortex with a pattern of BOLD activation that was significantly correlated to the time-series signal extracted from V5/MT on the right side of the brain when the monkey was shown the moving dot stimulus. The nature of the analysis ensured that V5/MT in the right hemisphere must have a significant correlation, but clearly the two hemispheres had similar connectivity as the pattern of activation we found was bilateral. This would be expected since both sides of the brain received the same visual stimulation. In addition to the STS, a large swathe of dorsal extrastriate cortex and parietal areas showed significant correlation, both in the control monkeys and in monkey S. Finally, the network connected to V5/MT included the Frontal Eye Fields (FEF) in monkey S and all control monkeys.

This connectivity analysis was also performed using the LGN and the pulvinar as a seed. Neither of these seeds produced a consistent pattern of connectivity with any cortical structures either in the control monkeys or monkey S.

## Structural subcortical connectivity of area V5/MT

The structural data indicated that both V5/MT and LGN were intact in monkey S, and fMRI activation data suggested that LGN and pulvinar were activated by a flickering checkerboard and moving dots respectively. Furthermore, extrastriate visual area V5/MT appeared to have a similar functional connectivity profile to this area as observed in control monkeys. The cortical activation evoked by visual stimulation was variable in all the monkeys, so to investigate the connection between V5/MT and visual subcortical structures, we performed probabilistic tractography on diffusion-weighted images. *Figure 9* shows the tracts between LGN and V5/MT and pulvinar and V5/MT, in each case with the threshold set at 10% of the maximum number of tracts reaching the target structure. In the data shown, the seed structure was either LGN or pulvinar and the target was ipsilateral V5/MT. The six control monkeys showed reasonably consistent tracts with pulvinar<->V5/MT tract generally running superior to the LGN<->V5/MT tract. Running the tracts in the other direction with V5/MT as the seed region produced comparable results. Monkey S also showed tracts between these structures, but the location and extent of the lesion meant that the actual route followed was different and the tracts seemed to be less direct as they appeared to project around the lesion.

The images of the tracts only give the route taken by the path, and do not allow comparison of tract strength or integrity. In order to quantify how the tracts in monkey S compare with the control animals, we performed two additional analyses. Firstly, we calculated the percentage of tracts terminating in the target structure. This gives an indication of the size of the tract, though this is not a direct measure of real pathway size. Secondly, we extracted the fractional anisotropy (FA) from each of the tracts independently. These metrics are shown in *Figure 10*, and indicate that the microstructure of the tracts in monkey S was most likely intact. But the number of tracts that could be identified between V5/MT and either LGN or pulvinar was very low, with the percentage of tracts reaching target ranging between 0.01–0.03% in all cases for monkey S. Compared to control monkeys, this was at least an order of magnitude lower.

Given the significantly weakened tract between V5/MT and subcortical nuclei indicated by the diffusion imaging, there is little evidence to suggest that a strengthening of these connections underlies whatever residual vision was present in monkey S.

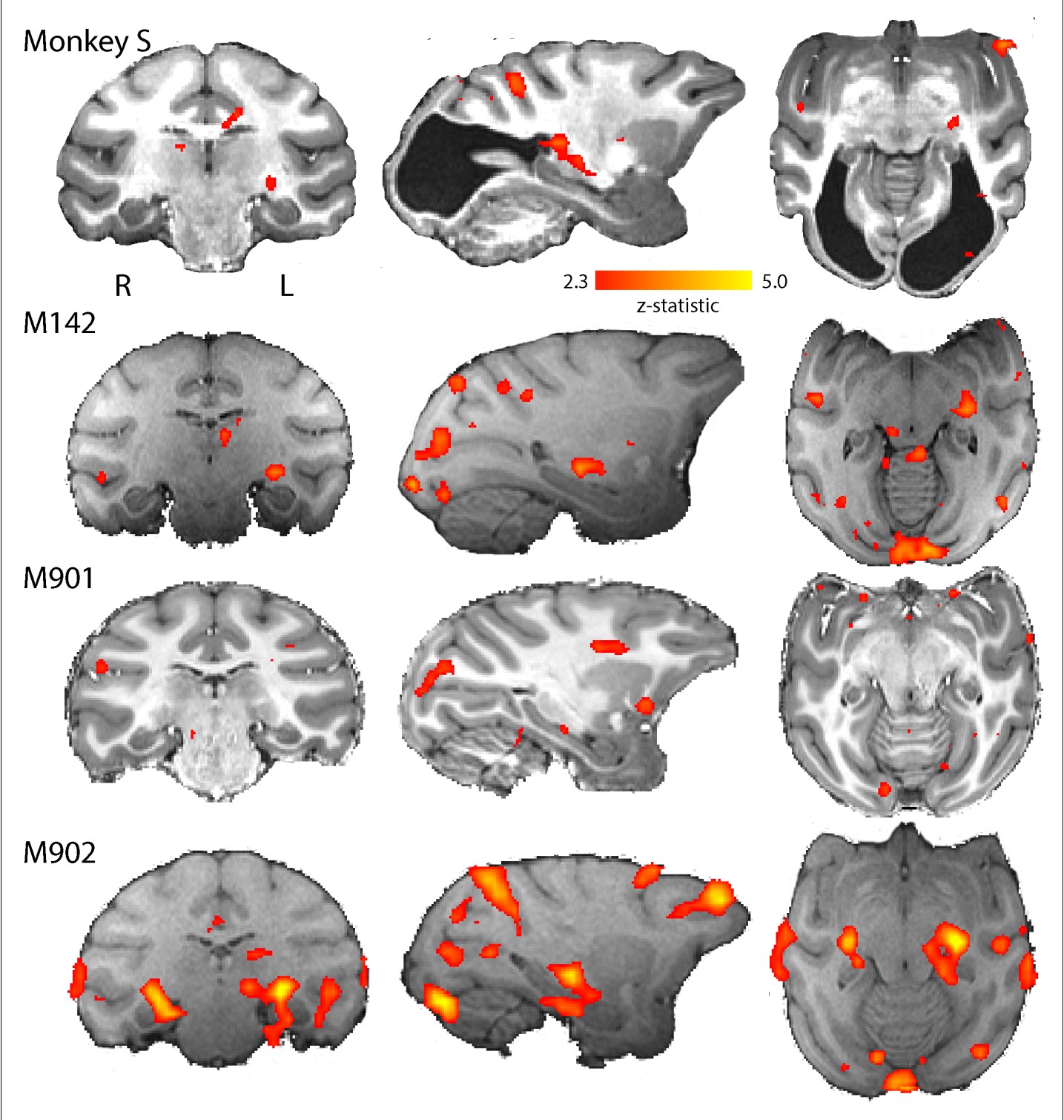

**Figure 5.** BOLD activation by visual motion. Activation to the moving dot stimulus generated less activity in all monkeys than the flickering checkerboard. In particular, the LGN activation levels were lower in all monkeys. Monkey S showed LGN activation in the left hemisphere only, but in this case pulvinar activation was also evident (z > 2.3).

DOI: https://doi.org/10.7554/eLife.42325.010

The following figure supplement is available for figure 5:

**Figure supplement 1.** BOLD activation by visual motion for monkey S.

*Figure 5 continued on next page*

*Figure 5 continued*

DOI: https://doi.org/10.7554/eLife.42325.011

## Cortical responses to face stimuli are present in monkey S

Residual vision can manifest in several different ways, including the ability to determine information from faces, as shown in patient TN (*Burra et al., 2013*), a function that has been suggested to survive loss of V1. To investigate whether any responses to faces could be detected in monkey S and two of the control monkeys, we presented full field stimuli of monkey faces. Blocks of neutral and threatening faces were interleaved with a blank screen. When we compared the BOLD response to all face stimuli compared to a mid-grey background (similar to the localizer used by *Liu et al., 2015*), we found clearly defined clusters of activation for the two most commonly identified temporal lobe face patches in the Rhesus macaque: the anterior (likely AF, Tsao et al. (2008)) and middle (likely MF, Tsao et al. (2008)) face patch located in the fundus of ventral STS (*Figure 11*). One control showed only the anterior face patch. While the locations of these areas are as previously described in macaques (*Tsao et al., 2003*; *Pinsk et al., 2005*; *Tsao et al., 2008*; *Bell et al., 2009*), we did not check the contrast against scrambled faces. We did not observe any specific differences in activation between threatening and neutral facial expressions in the STS, consistent with previous studies (*Hoffman et al., 2007*; *Hadj-Bouziane et al., 2008*).

## Discussion

Here, we have considered the visual activation and connectivity in a monkey with naturally-occurring bilateral damage to the visual cortex. The discovery of this monkey allowed a non-invasive approach to understanding the consequences of this damage to the structure and function of the visual brain similar to one that might be taken in human patients. In spite of the large occipital lesion, the structure and function of the remaining visual brain appeared largely unaltered. There are several

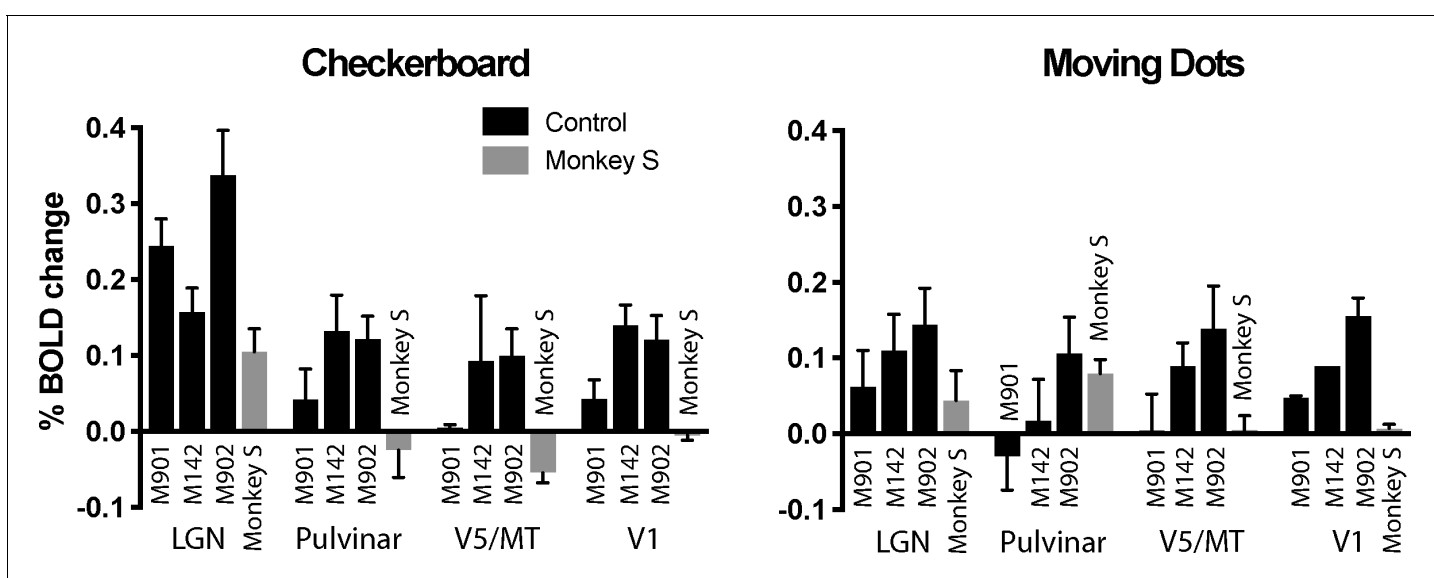

**Figure 6.** % BOLD change to visual stimulation. Change in BOLD signal to checkerboard and moving dot stimuli. In response to the flickering checkerboard stimulus, we saw robust BOLD activation of the LGN in monkey S and the three control monkeys. The activation level to the moving dots was lower in the LGN, although monkey S showed clear activation in the pulvinar only in response to moving dots. V1 was activated in all three control monkeys but not monkey S. Plotted are mean ± SEM of the right and left hemisphere for each monkey.

DOI: https://doi.org/10.7554/eLife.42325.012

The following figure supplement is available for figure 6:

**Figure supplement 1.** The mean timeseries of the BOLD response for monkey S averaged across the 16 cycles of one scan.

DOI: https://doi.org/10.7554/eLife.42325.013

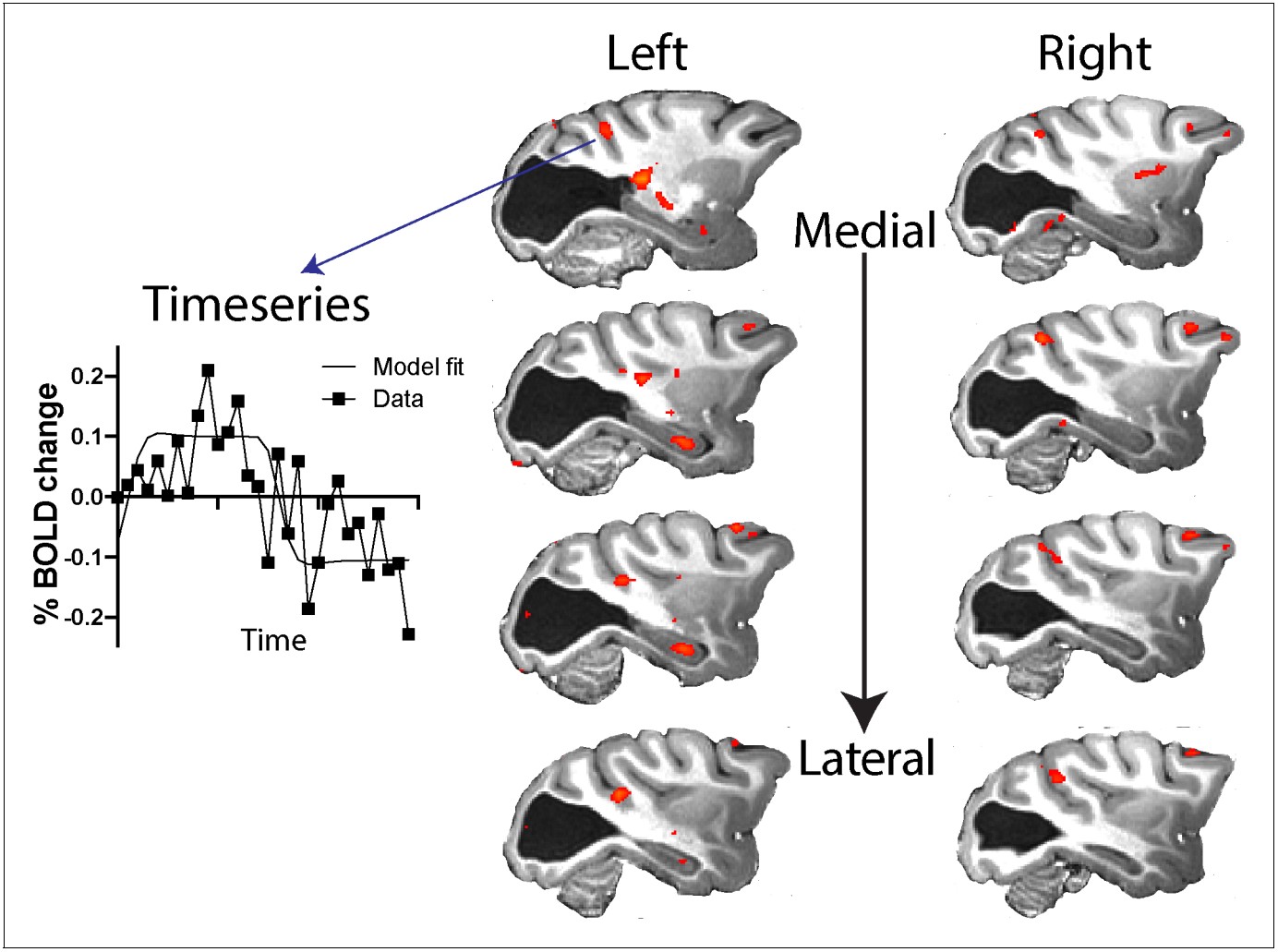

**Figure 7.** Response to visual motion in dorsal STS. Illustration of BOLD activation (z > 2.3) in series of slices through V5/MT in both hemispheres of monkey S. While there was no extended area of activation to the moving dot stimulus, there were a number of small regions of activation across dorsal regions of the STS. The timeseries plot shows the average signal extracted from one of these regions of activation (z > 2.3) shown in the top left panel.
DOI: https://doi.org/10.7554/eLife.42325.014

parallels in behaviour between children with cortical visual impairment (CVI) and this monkey. Children with CVI can easily be misdiagnosed and their behaviour can manifest as a variety of other conditions, including learning disability, without careful investigation (reviewed in *Chokron and Dutton, 2016*). Without access to imaging it would not have been possible to determine the loss of tissue in the occipital lobes in this animal.

## Visual structures outside of the lesion appeared structurally normal, although thalamocortical connections appeared to be reduced

The majority of literature relating to residual visual ability following damage to V1 describes adult-onset, unilateral damage due to stroke or trauma (e.g. *Cowey, 2010*). In these cases, the comparison between the intact and damaged hemispheres appears to show that the LGN and optic tracts are atrophied (*Bridge et al., 2011*). There are far fewer cases of peri-natal damage, but *Millington et al. (2014)* showed that even when the damage to occipital cortex is congenital, there is reduction of optic tract size in the affected hemisphere, implying a reduction in LGN size. A recent imaging study of a patient with bilateral occipital damage suggested atrophy of the LGN bilaterally (*Arcaro et al., 2018*). *Giaschi et al. (2003)* described a young man with bilateral damage to the occipital cortex suffered at birth. hMT+ in that case also appeared to be structurally normal, but did

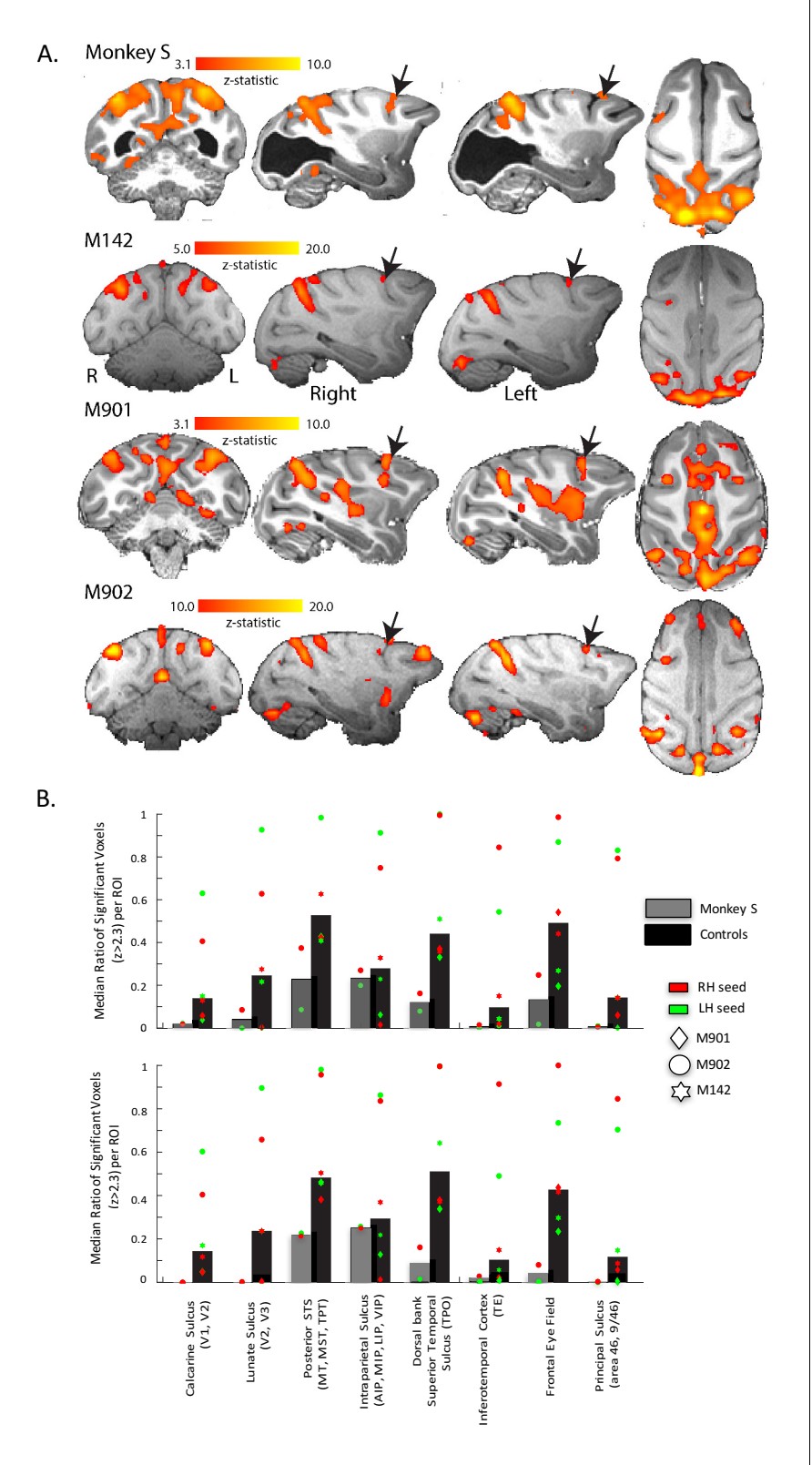

**Figure 8.** Functional connectivity of V5/MT. (**A**) Regions showing significant correlation with the time-series extracted from area V5/MT in the right hemisphere. The colourbar indicates the significance of the correlations (z-statistic) but note that the scales differ across monkeys. The network of cortical regions showing significant correlation is consistent across all animals, including monkey S, and consists predominantly of dorsal occipital and parietal regions as well as area FEF (black arrows). (**B**) Quantitative comparison of activation (z-statistic) of key visual, parietal and frontal brain regions.

*Figure 8 continued on next page*

*Figure 8 continued*

Upper bar graph shows ipsilateral and bottom contralateral connectivity expressed as the median ratio of significantly activated voxels over total number of voxels within each region of interest. Dots depict results separately for monkeys and seed regions. Despite individual differences in the level of cortical activation, a bilateral pattern of dorsal but not ventral visual activation emerged.

DOI: https://doi.org/10.7554/eLife.42325.015

not show evidence of activation to visual stimulation. It was not possible to determine whether the LGN was intact from the images provided in that paper.

Despite the large cortical lesion, in contrast to much of the human 'Blindsight' literature, we found the LGN in monkey S to be intact. Combined lesion and silencing studies in macaques reinforce the requirement for an intact LGN to support remaining visual function after V1 lesions (*Schmid et al., 2010*). Preservation of structure and function of the LGN after a visual cortical lesion might depend on age at the time of lesion as well as the extent of the lesion, as studies in marmosets suggest (*Atapour et al., 2017*; *Yu et al., 2018*; *Hendrickson et al., 2015*). This would be consistent with a very early loss of V1, possibly in utero, for monkey S.

The increased resolution of the structural scanning and consistent location of V5/MT in the macaque (*Zeki, 1974*; *Van Essen et al., 1981*) allowed the myelination of this area to be identified in vivo. Even with such extensive loss in the occipital lobe and thus, the loss of a major input (*Maunsell and van Essen, 1983*), this region of the STS appeared normal in monkey S. Conversely, the variability of hMT+ location in humans (*Large et al., 2016*) makes analysis of the myelin more challenging, and thus limits our opportunities to compare across the two species.

In spite of the apparent structural integrity of the LGN and V5/MT, the tracts between these regions appeared to be weaker, both in terms of the number of streamlines identified between them and the necessity for a more circuitous route to avoid the lesion. The circuitous route also points to an early, developmental origin for the lesion. Previous work in adult-acquired hemianopia in humans has indicated that tracts between LGN and hMT+ are similar in microstructure to sighted tracts in those showing some form of residual vision (*Bridge et al., 2008*; *Ajina et al., 2015b*). However, monkey S had very large bilateral lesions that may have caused greater disruption to the normal pattern of white matter connectivity. Nonetheless, the microstructure of these tracts appeared to be close to control values, as was also the case in previous human studies, so the reduction in size could reflect a reduction in feedback connections into the LGN rather than a change in feedforward connectivity into cortex.

## Limitations of analysing visual activation under general anaesthesia

Performing BOLD fMRI in anaesthetised macaques is challenging even when the monkey has a healthy visual system; this is evident in the variability in the amount of cortical activation in the control monkeys. The level of BOLD activation is affected by general anaesthesia parameters such as type and depth (*Hutchison et al., 2014*; *Vincent et al., 2007*), although previous results do not explain the potential differences in dorsal and temporal visual activations we see. We used isoflurane anaesthetic agent, under which it is known that some visual cortical activation is preserved, at least at low doses (*Logothetis et al., 1999*; *Vincent et al., 2007*; *Goense and Logothetis, 2008*). We mitigated some of this effect, by restricting our quantitative analyses to those sessions that showed significant activation of the LGN with a checkerboard stimulus. However, we cannot exclude the possibility that some of the variability in cortical activity, specifically for area V5/MT, is due to effects of the anaesthetics which vary at shorter time scales within a fMRI session. It has also been suggested that some variability in cortical activation and functional connectivity as measured with fMRI could be linked to inter-individual differences in the morphology of sulcal patterns (*Lopez-Persem et al., 2019*; *Xu et al., 2019*). In the same context, some of the small patches of activations that we see around dorsal and ventral STS in monkey S could be spurious signal fluctuations. However, for monkey S, looking through the whole brain (see *Figure 4—figure supplement 2*, *Figure 5—figure supplement 1*), we find only few regions with significant activation. A further challenge in monkey S is that the damage is bilateral. In much of the previous work in humans, the sighted hemisphere has been used as a control area, therefore controlling for any global variation in activation levels.

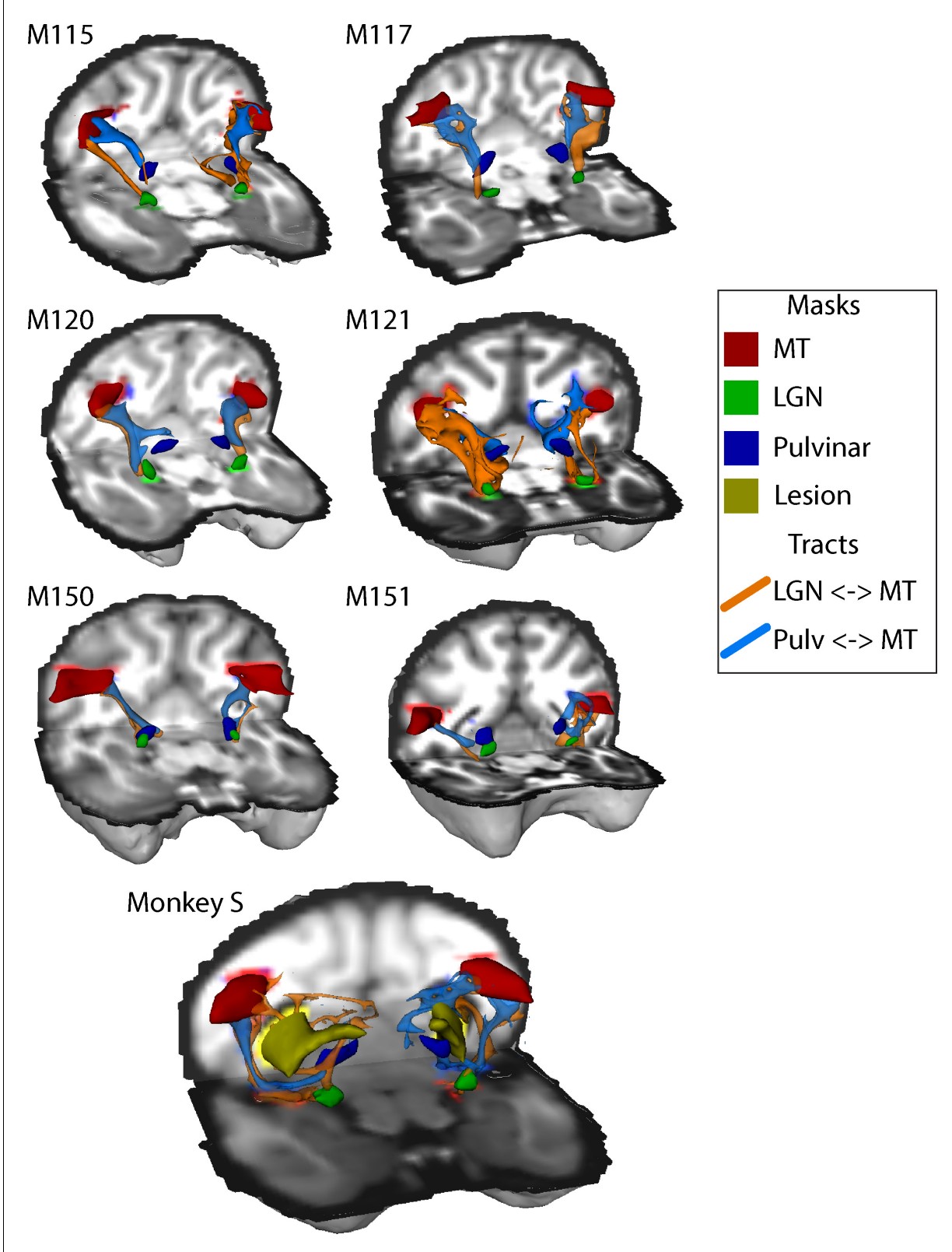

**Figure 9.** White-matter tracts between LGN, pulvinar and V5/MT. Diffusion-weighted imaging and probabilistic tractography were used to investigate tracts between extra striate visual motion area V5/MT (red) and the LGN (green) and V5/MT and the pulvinar (blue). All tracts could be traced in the control monkeys and monkey S. In monkey S, the tracts appeared fragmented and took different routes, likely due to the presence of the lesion (shown in yellow).

*Figure 9 continued on next page*

*Figure 9 continued*

DOI: https://doi.org/10.7554/eLife.42325.016

## Local activation patterns are preserved

The most evident activation is clearly in the LGN in response to checkerboard stimulation, which is equivalent to that seen in controls and suggests the visual pathway prior to V1 is intact. The activation level to moving dots was considerably lower, but that would be predicted from the properties of LGN cells. By contrast, the activation pattern in the pulvinar to moving dots for monkey S appeared to be as great as in the best control subject, which is consistent with an increased role for this structure due to the perinatal nature of the lesion (*Warner et al., 2015*).

Moving dot stimuli in the healthy and hemianopic human brain consistently lead to activation of hMT+, even when damage is bilateral (*Arcaro et al., 2018*; *Bridge et al., 2010*). Thus, the lack of consistent activation in V5/MT in monkey S and some controls suggests a suppressive effect of the anaesthetic, which was most evident with larger doses of the volatile anaesthetic. Also, drifting eye movements and inappropriate ocular accommodation under anaesthesia could have affected the functional activation we measured. Nevertheless, for visual motion stimuli we saw in monkey S foci of activation in the region of V5/MT in dorsal STS of both hemispheres, suggesting any activation may be sub-threshold and difficult to detect with BOLD.

Even without the strong visually-evoked V5/MT responses, it was possible to map out a functional network of cortical areas showing a similar BOLD response over time to V5/MT. This was very similar in monkey S and the control monkeys, suggesting that this inter-cortical activity in the macaque visual brain was not affected by the loss of V1. Interestingly, none of the monkeys included LGN in this functional network, perhaps reflecting the weak anatomical connectivity between these areas in

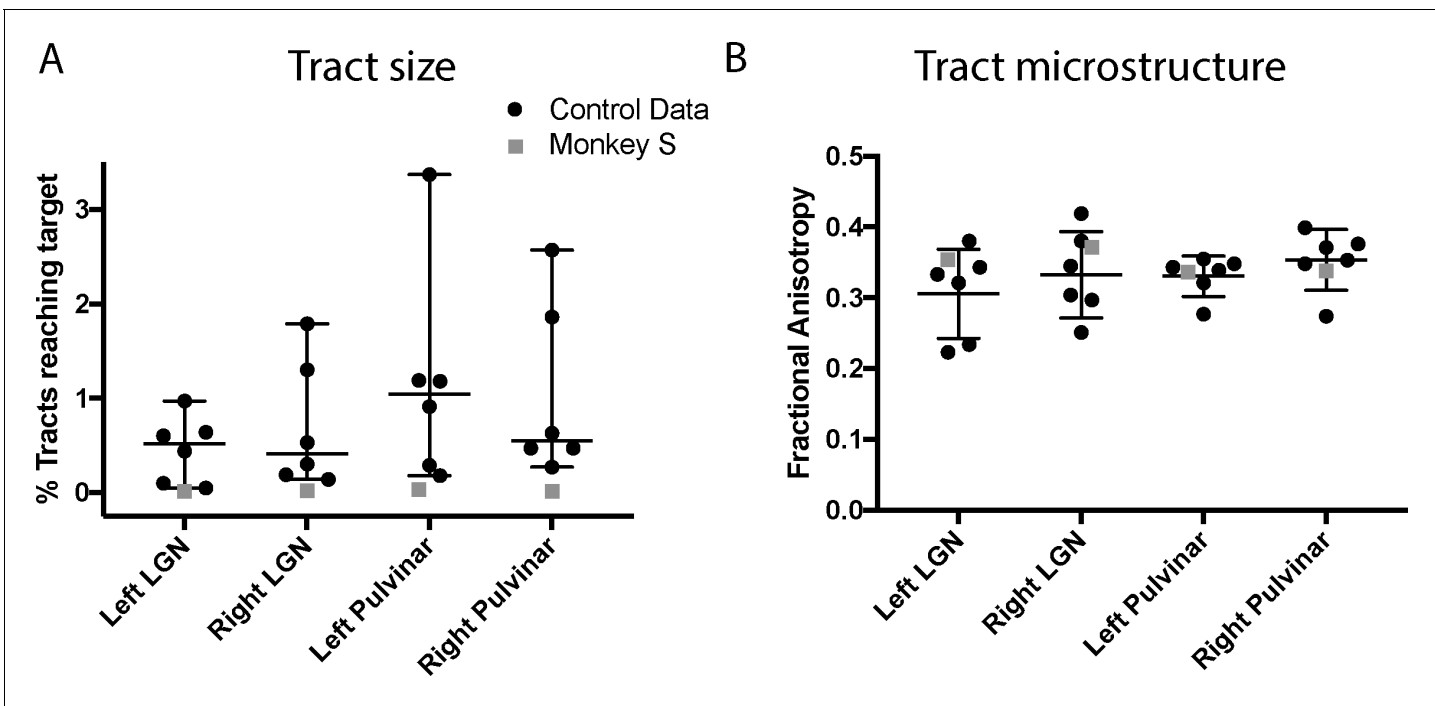

**Figure 10.** Quantification of tractography results. (**A**) Quantification of the percentage of streamlines from the seeds in subcortical areas LGN and pulvinar reaching cortical area V5/MT in the two hemispheres. While there was considerable variability between control monkeys (black circles), the number of tracts in monkey S (grey square) is consistently lower for all tracts. (**B**) By contrast, the functional anisotropy values for the tracts in monkey S were comparable to the control values for both pathways. This suggests that the white matter microstructure within the tracts between V5/MT and each target structure was intact. Data points show results from individual hemispheres, the horizontal lines give the median and the 95% confidence interval for the control group.

DOI: https://doi.org/10.7554/eLife.42325.017

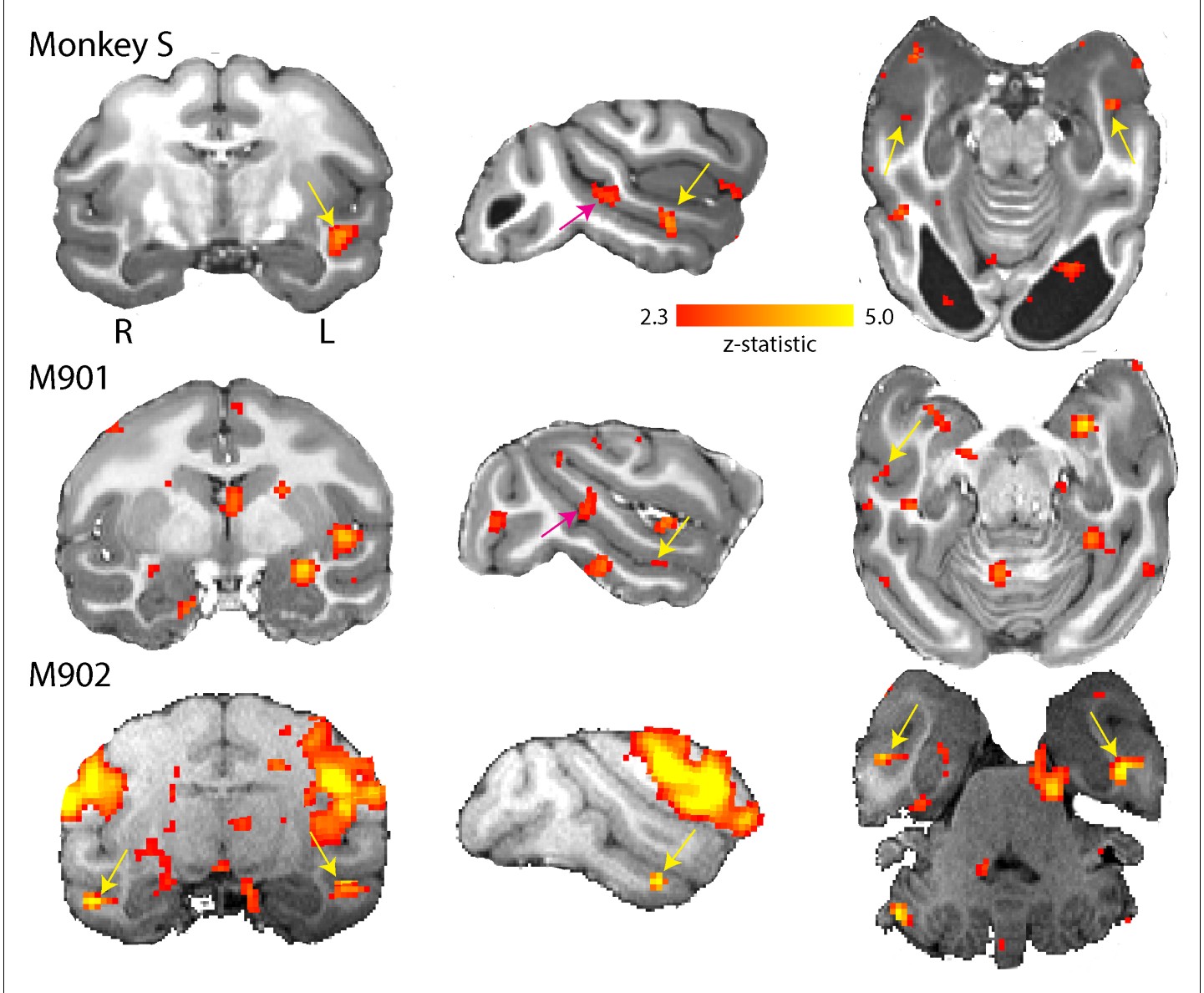

**Figure 11.** Activation of anterior and middle face patch for face stimuli. Images show the BOLD signal for blocks of neutral and threatening faces and compared to a mid-grey screen. Monkey S showed activation of the middle fundus (pink arrow) and anterior fundus (yellow arrow) face patch along the mid- and temporal STS regions. Both controls showed activation of the anterior face patch and one also showed activation of the middle face patch. (2.3 < zstat < 5).

DOI: https://doi.org/10.7554/eLife.42325.018

typical animals (*Sincich et al., 2004*). For all monkeys, FEF was included in the functional network, as would be predicted from a knowledge of the V5/MT connectivity (*Schall et al., 1995*).

Surprisingly, we also found activation of two of the most pertinent face patches along the STS in monkey S – in the context of little other consistent activations. Without the comparison to scrambled faces, this result needs further scrutiny. However, if confirmed, this result would suggest other intact subcortical inputs in this monkey to temporal visual cortex from pulvinar, claustrum, and amygdala supporting these activations (*Grimaldi et al., 2016*).

### Likely pathway for supporting residual visual function

Given that the day-to-day behaviour of monkey S did not cause concern, the monkey presumably had a reasonable amount of residual vision, in spite of the extensive cortical lesions. Considering all the evidence, it appears that the dorsal visual network was intact, and would have the potential to support active vision (*Koyama et al., 2004*; *Davare et al., 2011*), as seen in a number of human patients with damage to the visual cortical system (*Goodale and Milner, 1992*; *Bridge et al., 2013*; *Goodale, 2011*). Retinal information may reach the dorsal visual areas via the weak pathways with LGN and pulvinar, but the lack of consistent visual cortex activation, possibly due to the anaesthesia, makes this question difficult to answer. The lack of significant differences between monkey S and controls in the pathways between LGN-V5/MT and pulvinar-V5/MT suggests either or both could support residual visual abilities.

So, what might be the source of visual input that supports visual function and presumably shaped an intact extrastriate visual brain network? One possibility could be some preserved V1 connectivity. But given the size and location of the bilateral lesion, this argument is difficult to sustain. Another possibility would parallel, alternative inputs to cortex. Earlier MRI studies in V1-lesioned monkeys have shown parallel activation in early visual areas, like V2, V3 and V4 as well as V5/MT (*Schmid et al., 2010*). It has been suggested that V2 might be a crucial contribution to visual function and awareness (*Merigan et al., 1993*). These visual cortical areas also receive direct subcortical input (*Wong-Riley, 1976*; *Kaas and Lyon, 2007*). Based on our knowledge of functional representations, the activation of face patches in temporal cortex discussed above must be driven by a visual subcortical input other than to V5/MT. Since it is likely to be a wider pattern of cortical activation that underpins visual awareness and behaviour (e.g. *Papanikolaou et al., 2019*), it might be supported by a wider pattern of weak, parallel subcortical inputs directly to extrastriate visual cortex.

### Conclusions

Visual structures, both cortical and subcortical, outside of the large lesion of primary visual cortex remained intact in monkey S with little evidence of atrophy. This suggests a role in residual visual function, consistent with an intact network of extrastriate cortical visual areas that was comparable to control monkeys. While the structural connectivity between the subcortical regions and area V5/MT was weak, the microstructure was intact. Thus, unlike adult-acquired lesions, there appeared to be a maintenance of structural integrity of the visual system when V1 is damaged neonatally. This may explain the increased residual function both, in monkey S and in children with early damage to the visual cortex.

## Materials and methods

### Animals

Seven macaque monkeys (*Macaca mulatta*, two female and five male), weighing 6.25 to 12.3 kg (mean weight ± SE: 9.1 kg ± 0.8) underwent functional and structural MRI scans under general anaesthesia. Previously collected anatomical data from a further 10 macaque monkeys (four females for myelin/structural scans; six males for diffusion-weighted imaging (DWI)/structural scans; weighing 4.35 kg to 11 kg) were analysed from MRI scans (MPRAGE, T2w, DWI), also obtained under general anaesthesia. The structural myelin data of the four control monkeys have previously been reported elsewhere (*Large et al., 2016*), and so have the DWI data from four of the six controls (*Rafal et al., 2015*). The monkeys were socially housed together in same sex groups of between 2 and 6 animals and housing and husbandry were in compliance with the ARRIVE guidelines of the European Directive (2010/63/EU) for the care and use of laboratory animals. All animal procedures were carried out in accordance with Home Office (UK) Regulations and European Union guidelines (EU directive 86/609/EEC; EU Directive 2010/63/EU).

### Behavioural training and tasks

Transport box training and primate chair training protocols employed here were recently described elsewhere (*Mason et al., 2019*). For task training, monkey S was usually trained once a day for five days per week (typically 30 min – 105 min per day). But for double transport device acclimatization (transition from transport box to primate chair), monkey S was trained twice a day for five days per

week. Task training in the transport box took place in the morning and acclimatization to enter a primate chair in the afternoon.

For touchscreen training, monkey S was brought in the transport box from the home enclosure to a test cubicle (as previously described in *Mitchell et al., 2007*). In brief, the transport box was fixed to a touch-sensitive video screen (380 × 280 mm, 800 × 600 pixel). Stimulus presentation, recording of screen touches, and reward delivery were computer-controlled. For a correct touch, a dispenser delivered 190 mg banana-flavoured pellets accompanied by a click. In this training environment, monkeys progress through increasing the number of completed trials day-by-day and following a set of tasks of increasing difficulty. The first two simple tasks in this sequence were attempted by monkey S: (i) In 'anytouch', monkeys were rewarded for touching the screen anywhere during presentation of an array of colourful, alphanumeric characters (length up to 34 mm) on a black background; (ii) in 'oneplace', a single coloured typographical character was presented on a black background in a random location and remained on the screen until the monkey had touched it. A touch caused the character to disappear and a reward pellet to be delivered.

Working with the touch screen while seated upright in a primate chair was another training set. Once acclimated to the primate chair, reward pellets were re-placed with banana smoothie taken from a spout and delivered by a reward pump. The primate chair was positioned directly in front of the touchscreen monitor (as described above). Monkey S's head was approximately 160 mm away from the touchscreen and her hand reach 100 mm. Tasks were the same as described above for the transport box.

Further training switched to a different effector, using a paddle lever touch sensor attached to the front of the primate chair. This task required monkey S to press a lever for less than 500 msec, while a red circle (5 mm in diameter, approx. 1.8° of visual angle) was displayed in the centre of the touchscreen and to release the lever when the circle switched to green. In addition, monkey S had to reach out to touch one of two white squares (50 mm in size, approx. 18° of visual angle) that were then presented on the touchscreen to the left and right of the circle.

In another version of the task, the lever was replaced with a metal knob. Here, the training program displayed a red circle (diameter = 40 mm, approx. 14° of visual angle) on the touchscreen, directly above the knob. Touching the metal knob for 200 msec delivered the reward and the red circle disappeared for a 2 s inter–trial interval (ITI). Monkey S was initially trained to hold a metal bolt and her hand was guided toward the metal knob.

During training, different fluid rewards (e.g. different banana smoothie concentrations, blackcurrant juice, apple juice) were used to determine preferences. For the later part of the task training, monkey S was on a fluid control protocol to increase motivation: water bottles were removed each night before the next day's training session and replaced after each daily training session.

## Anaesthesia

The seven monkeys undergoing functional MRI scans were sedated with a mixture of ketamine (7.5 mg/kg), xylazine (0.125 mg/kg) and medetomidine (0.1 mg/kg). They were intubated, an i.v. cannula was inserted into the saphenous vein for fluid infusion (Hartmann's solution, 2 ml/kg/hr) and non-invasive BP, rectal temperature, heart rate and oxygen saturation were continuously monitored. They were placed in an MRI compatible stereotaxic frame with anaesthetic cream (EMLA cream) applied to pressure points, and 'viscotears' applied to stop the eyes from drying. During the scan, they were ventilated with a gaseous mixture of isoflurane in oxygen (range 0.8% to 2.0%) with end-tidal $CO_2$ maintained around 38 mmHg. Between scan sequences approximately hourly, legs were massaged and 'viscotears' were re-applied. The isoflurane anaesthetic gas mixture during scanning sequences was kept to a level commensurate with adequate anaesthesia through monitoring of physiological parameters (heart rate, end-tidal $CO_2$, blood pressure) and, between sequences, through pinch test. In the majority of monkeys during visual presentation, isoflurane concentration was at 1% (range: 0.8% to 1.6%). In anaesthetised monkeys, visual activation can vary for a number of reasons, including accommodation, drifting eye movements and level of anaesthesia. In our data, there was some indication that lower average levels of isoflurane were associated with significant visual stimulation responses, but the pattern was variable (see *Figure 4—figure supplement 1*). Procedures for monkeys undergoing structural scans are described elsewhere (*Large et al., 2016*; *Rafal et al., 2015*).

## Scanning sequences

Anesthetised monkeys were placed in an MRI-compatible stereotactic frame (Crist Instrument) in sphinx position. For visual presentations, eyelids were held open with surgical tape. Data were acquired with a 3T clinical MRI scanner, using a four-channel phased-array radiofrequency coil in conjunction with a local transmission coil (H. Kolster, Windmiller Kolster Scientific, Fresno, CA).

Five high-resolution (voxel size 0.5 mm x 0.5 mm x 0.5 mm, TE = 4.04 ms; TR = 2500 ms; flip angle = 8°, 128 slices) T1-weighted structural images were acquired using a 3D magnetisation-prepared rapid-acquisition gradient echo (MPRAGE) sequence. To compute myelin-weighted images, we also acquired 13 T2w 3D turbo spin-echo (TSE) scans with variable flip angle (T2w; voxel size $0.5 \times 0.5 \times 0.5$ mm, TE = 3.51 ms, TR = 100 ms, flip angle = 45°, 128 slices) within the same session. Scans of the same type were averaged for each monkey; the mean image of the T1w MPRAGE scans was then divided by the mean image of the T2w scans to create a T1w/T2w image, which we refer to as a T1w/T2 w 'myelin-weighted map' (*Glasser and Van Essen, 2011*; *Large et al., 2016*).

We acquired the diffusion-weighted imaging (DWI) data with a twice-refocused spin-echo (TRSE) sequence. The DWI dataset included six b = 0 s/mm$^2$ and single shell with 60 gradient directions using b = 1000 s/mm$^2$. Whole-brain DWI volumes were collected at 1 mm x 1 mm x 1 mm resolution (FOV = 112 mm x 112 mm, image matrix $112 \times 112$) as 56 interleaved axial slices. For monkey S, TE = 102 ms and TR = 9 s; for the controls, TE = 102 ms and TR = 8.3 s. Each 60-direction, diffusion-weighted imaging (DWI) scan took 13 min, and was repeated 12 times in each monkey for subsequent averaging to improve SNR. Alternate sets of diffusion-weighted data were collected with the phase encode direction was reversed (for monkey S, right-left and left-right reversal; for the controls, anterior-posterior and posterior-anterior reversal), so that six sets of each direction were collected. Alternating phase-encoded images for each animal were later combined to reduce susceptibility artefacts along the phase-encoding direction using 'Top-Up' (*Andersson et al., 2003*; *Smith et al., 2004*).

fMRI data were acquired using a gradient-echo T2* echo planar imaging (EPI) sequence with 1.5 mm x 1.5 mm x 1.5 mm resolution, 32 ascending slices, TR = 2.00 s, TE = 29 ms, flip angle = 78. A block was 30 s long, with stimulus and baseline conditions interleaved. We collected at least 32 min of functional data from each subject and each stimulus condition.

## Post mortem scans

Prior to scanning, the *post mortem* brains were removed from 4% paraformaldehyde and placed into phosphate-buffered saline for at least 72 hr, then they were placed in a sealed container into 3SM Fluorinert Electronic Liquid FC3283 (Acota Ltd). *Post mortem* brains were scanned on a whole-body Siemens 7T MRI scanner (28 channel knee coil - QED), where T2*-weighted images were acquired with seven 3D gradient-echo (GRE) scans (voxel size 0.27 mm x 0.27 mm x 0.27 mm, TE = 18 ms, TR = 38 ms, flip angle = 15°, 256 slices). A rigid body (translation and rotation) co-registration was performed between repeats using FSL FLIRT (*Jenkinson and Smith, 2001*; *Jenkinson et al., 2002*) prior to averaging and a Gibbs ringing correction was carried out on the combined datasets (*Kellner et al., 2016*).

## Visual stimuli for functional MRI

Visual presentations of high contrast stimuli (black:white ca. 500:1) were back-projected onto a screen (with resolution of $1280 \times 1024$ pixels subtending a visual field of 105° x 75°). The screen was placed at a distance of 19 cm centrally in front of the opened eyes of the monkey positioned in the sphinx-position in the scanner. Monkey S and six fMRI control monkeys were presented with checkerboard and visual motion stimuli (of which monkey S and three controls showed visual responses in LGN to the checkerboard). Only Monkey S and four control animals were presented with the face stimuli.

*Checkerboard.* The checkerboard stimulus was created from two inverted circular stimuli each divided into wedges with alternating contrasts (*Figure 12A*). Both stimuli had a radial width of 320 by 256 pixels and angular distance of 45 degrees (or radial frequency = 2 cycles and angular frequency = 8 cycles). Stimuli alternated at 1.6 Hz. The flickering checkerboard alternated with a mid-grey screen with a block length of 30 s. Each scan consisted of 16 repeats of this 60 s cycle, giving a length of 480 volumes. Two scans were run in the session.

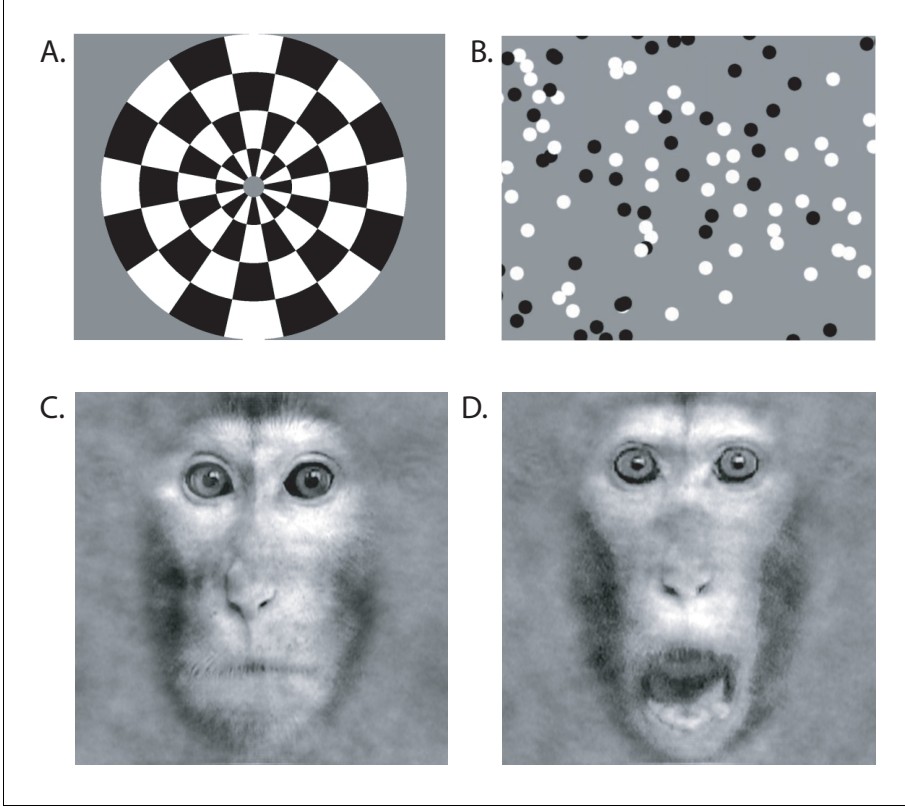

**Figure 12.** Visual Stimuli for functional MRI. (**A**) Contrast reversing checkerboard was used for eliciting basic visual activations (one of two images shown). (**B**) For eliciting motion-related visual responses, black and white random dots were shown full screen on a mid-grey background moving coherently in one of the four cardinal direction. (**C**) Face stimuli were either neutral or (**D**) threatening.
DOI: https://doi.org/10.7554/eLife.42325.019

## Visual motion

To identify areas responding to visual motion, we showed a full field of 100% coherently moving white and black dots on a mid-grey background (*Figure 12B*): 7.5 s for each of four directions (0°, 90°,180°, 270°, pseudorandomized). Each dot measured 20 × 20 pixels (about 1.6° x 1.6°). The baseline stimulus was a 30 s display of stationary dots. Block and scan length were the same as for the checkerboard stimulation.

## Faces

Face stimuli were greyscale images depicting the frontal view of the faces of three individual monkeys (stimuli provided by *Inagaki and Fujita, 2011*) (*Figure 12C–D*). Stimuli were presented centrally and subtended 33° of visual angle. Faces had either a neutral expression ('neutral block', mouth closed) or a threatening expression ('fear block', mouth open, teeth showing). Stimuli were presented in 30 s blocks containing random repetitions of the relevant images with a 16 s inter-block interval. Within each block the presentation of each stimulus was presented for 0.8 s with an inter-image interval of 1 s. A third 30 s block consisted of a grey background screen.

## Definition of visual area masks

Masks for the lateral geniculate nucleus (LGN), medial and lateral pulvinar, superior colliculus (SC), area V5/MT and primary visual cortex (V1) were defined anatomically on the brains of each individual monkey in structural space using a standard atlas as a guide (National Institute of Mental Health Macaque Template - NMT) (*Seidlitz et al., 2018*).

## Analysis of functional MRI data

To control for the effects of anaesthesia, we included four monkeys for analysis of functional MRI data which showed significant LGN activation to the checkerboard stimulation (z > 2.3, not further corrected) (See *Figure 4—figure supplement 1*).

*Block experiment analysis.* The checkerboard and motion experiments followed the same analysis pipeline. In the checkerboard experiment, the flickering checkerboard was contrasted with a mid-grey screen, whereas in the motion experiment, moving dots were contrasted with stationary dots. Pre-processing and statistical analysis were performed using tools from the FSL toolbox (FMRIB Software Library, http://www.fmrib.ox.ac.uk/fsl). Non-brain tissue was excluded from analysis using BET (Brain Extraction Tool) (*Smith, 2002*), motion correction was performed using MCFLIRT (FMRIB Linear Image Restoration Tool with Motion Correction) (*Jenkinson et al., 2002*; *Jenkinson and Smith, 2001*). Spatial smoothing was applied using a full-width half-height Gaussian kernel of 3 mm and high pass temporal filtering (Gaussian-weighted least-squares straight-line fitting) was used. Functional images were registered to high-resolution structural scans using FLIRT.

A general linear model (GLM) was used to contrast the presentation of the checkerboard or moving dots against the mid-grey or stationary dot background and data from the two stimulus runs were combined using a fixed effects analysis. Statistical maps were thresholded at a z-statistic of 2.3, with no further correction. This relatively liberal threshold was chosen due to the anaesthesia reducing the BOLD signal in the monkeys, although our isoflurane levels were below those shown to induce significant resting network changes (*Hutchison et al., 2014*).

For the fMRI analyses, visual areas masks for LGN, pulvinar and V5/MT were transformed into EPI space and a region of interest analysis was performed by extracting the % BOLD change from each area using Featquery, another tool from the FSL toolbox.

Timeseries shown for monkey S were extracted from one of the two scans obtained for each stimulus, specifically the one with the highest z-statistics, because signal modulation was likely to be noisy. Data were averaged across the 16 stimulus cycles to give a mean cycle of the timeseries. In the case of the LGN and pulvinar timeseries, the data were averaged across all voxels in the mask, rather than using peak voxels. In the case of the STS, the timeseries data were extracted from the region of activation (z > 2.3) indicated by the arrow in the figure.

## Functional connectivity analysis

To determine whether the pattern of functional connectivity within the cerebral cortex in the healthy visual system was evident in monkey S, the BOLD time series was extracted from the V5/MT ROIs using the FSL function 'fslmeants'. The timeseries was then used as the explanatory variable in a FEAT analysis to identify the brain areas showing a significant statistical relationship with V5/MT. This was performed for both the checkerboard and motion scans, and a fixed effects analysis was used to combine the two repeats of each stimulus type. For the quantitative analysis, we defined broad regions of interest and counted the ratio of activated voxels to total voxels in this region.

## Faces

Data were analysed using statistical parametric mapping (SPM12, *Friston et al., 2006*). Pre-processing steps consisted of realignment and co-registration to the monkeys' own structural images. Images were smoothed using an isotropic Gaussian kernel (full-width at half maximum: 3 mm x 3 mm x 3 mm). Realignment parameters were included as covariates of no interest in the design. To localise face-related activations we calculated the contrast [Faces Vs Blank], and resulting z-scores were displayed on the monkeys' own structural images using FSLEyes (z-score >2.3).

## Analysis of DWI data and probabilistic tractography

Probabilistic tractography was performed using ProbtrackX2 from the FSL FDT toolbox (*Behrens et al., 2007*). We traced two unilateral pathways in each hemisphere: pulvinar to V5/MT and LGN to V5/MT. Masks for these three structures were obtained from a standard atlas (NMT) (*Seidlitz et al., 2018*) and later modified by hand. The pulvinar mask was reduced in size to focus on the inferior pulvinar as this is the portion that relays visual information from the SC to area V5/MT (*Berman and Wurtz, 2010*). Anatomical masks used for tractography were further modified to eliminate potential overlap between masks for neighbouring subcortical regions. For example, the LGN

and pulvinar masks were modified so that they were always separated by at least 1–2 coronal slices. We used exclusion masks to eliminate streamlines passing anterior of the LGN or across hemispheres. In the case of the V1-lesioned monkey S, we also included a mask encompassing the bilateral lesion.

We modified the default parameters of ProbtrackX2 in order to optimise the procedure for NHP data, based on previous work in our lab (*Tang-Wright, 2016*). Specifically, we limited the streamline length to 100 steps, with step length of 0.5 mm. The value of each voxel represented the total number of streamline passing through. Each voxel was thresholded at 10% of the maximum number of streamlines found in any voxel. A recent study that directly compared diffusion tractography with tracers in monkeys reported that a threshold of 10% most reliably reflect the anatomy when compared with tracers (*Azadbakht et al., 2015*).

## Acknowledgements

John Duncan for making this project possible. Mikio Inagaki for generously providing the facial expression stimuli. Funded by Wellcome Trust Strategic Award 101092/Z/13/Z (to AJP, MB, KK, AM), BBSRC grant BB/H016902/1 (to KK, HB, KLM), MRC grant MR/K014382/1 (to HB, AJP), BBSRC David Phillips Fellowship BB/N019814/1 (to RBM). Saad Jbabdi for advice about the analysis. BMS and veterinary staff for support with the anaesthetised MRI scans and animal care. Stuart Mason for involvement in the behavioural training.

## Additional information

### Funding

| Funder | Grant reference number | Author |
| --- | --- | --- |
| Wellcome | 101092/Z/13/Z | Andrew J Parker<br>Mark Buckley<br>Kristine Krug<br>Anna S Mitchell |
| Biotechnology and Biological Sciences Research Council | BB/H016902/1 | Kristine Krug<br>Holly Bridge<br>Karla L Miller |
| Medical Research Council | MR/K014382/1 | Holly Bridge<br>Andrew J Parker |
| Biotechnology and Biological Sciences Research Council | BB/N019814/1 | Rogier B Mars |

The funders had no role in study design, data collection and interpretation, or the decision to submit the work for publication.

### Author contributions

Holly Bridge, Conceptualization, Formal analysis, Supervision, Investigation, Visualization, Methodology, Writing—original draft, Writing—review and editing; Andrew H Bell, Formal analysis, Investigation, Writing—review and editing; Matthew Ainsworth, Investigation, Methodology, Writing—review and editing; Jerome Sallet, Elsie Premereur, Formal analysis, Investigation, Methodology, Writing—review and editing; Bashir Ahmed, Anna S Mitchell, Investigation, Writing—review and editing; Urs Schüffelgen, Data curation, Investigation; Mark Buckley, Resources, Supervision; Benjamin C Tendler, Investigation, Methodology; Karla L Miller, Supervision, Methodology; Rogier B Mars, Resources, Investigation, Methodology; Andrew J Parker, Resources, Funding acquisition, Investigation, Methodology, Writing—review and editing; Kristine Krug, Conceptualization, Resources, Data curation, Formal analysis, Supervision, Funding acquisition, Investigation, Methodology, Writing—original draft, Writing—review and editing

## Author ORCIDs

Holly Bridge (iD) https://orcid.org/0000-0002-8089-6198
Andrew H Bell (iD) https://orcid.org/0000-0001-8420-4622
Matthew Ainsworth (iD) http://orcid.org/0000-0002-5767-7704
Jerome Sallet (iD) http://orcid.org/0000-0002-7878-0209
Elsie Premereur (iD) https://orcid.org/0000-0002-5054-9515
Bashir Ahmed (iD) http://orcid.org/0000-0001-9989-2659
Anna S Mitchell (iD) https://orcid.org/0000-0001-8996-1067
Urs Schüffelgen (iD) https://orcid.org/0000-0002-0268-1650
Mark Buckley (iD) http://orcid.org/0000-0001-7455-8486
Benjamin C Tendler (iD) https://orcid.org/0000-0003-2095-8665
Karla L Miller (iD) https://orcid.org/0000-0002-2511-3189
Rogier B Mars (iD) https://orcid.org/0000-0001-6302-8631
Andrew J Parker (iD) http://orcid.org/0000-0001-5800-0407
Kristine Krug (iD) https://orcid.org/0000-0001-7119-9350

## Ethics

Animal experimentation: Ethics Statement: All procedures were performed in accordance with United Kingdom Home Office regulations and European Union guidelines on animal experimentation (EU directive 86/609/EEC; EU Directive 2010/63/EU). Animal protocols passed local and national ethical review and were licensed by the UK Home Office. MRI scans were conducted under general anaesthesia and procedures and care were regularly reviewed with veterinary staff to ensure best practise.

## Decision letter and Author response

Decision letter https://doi.org/10.7554/eLife.42325.022
Author response https://doi.org/10.7554/eLife.42325.023

## Additional files

### Supplementary files

• Transparent reporting form
DOI: https://doi.org/10.7554/eLife.42325.020

### Data availability

We are commited to sharing tools, data and results openly. There is still no general repository for NHP MRI images. Recently, some neuroimaging datasets collected in Oxford have been shared as part of the PRIME-DE initiative (Milham et al. 2018), but we understand there is no general mechanism for new deposits at present. We will make our own key source imaging data used for this study available through an online weblink: https://web.gin.g-node.org/hbridge_oxford/brainwithoutv1. Through this mechanism, we will share the structural and functional source MRI data we acquired. The controls for the diffusion-weighted imaging data are re-analysed from a previous paper, which we cite.

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
