## [Decision Letter]

Thank you for submitting your article "Intact extrastriate visual network without primary visual cortex in a Rhesus macaque with naturally occurring Blindsight" for consideration by *eLife*. Your article has been reviewed by two peer reviewers, and the evaluation has been overseen by a Reviewing Editor and Joshua Gold as the Senior Editor. The following individual involved in review of your submission has agreed to reveal his identity: Michael Schmid (Reviewer #1).

The reviewers have discussed the reviews with one another and the Reviewing Editor has drafted this decision to help you prepare a revised submission.

Summary:

Bridge et al. present a very interesting case report of monkey S with a likely congenital bilateral occipital lesion visible as enlarged ventricles in structural MRI. The monkey presents without an apparent visual deficit in the home cage. Six intact controls were also studied. The authors use MR based methods during anaesthesia to report functional activation and connectivity of thalamic and cortical regions. While BOLD fMRI signal modulation is observed in LGN and pulvinar, no such activation is observed at significance level in striate or extrastriate visual cortex (apart from face patches, interestingly). Further analysis reveals residual connectivity of area MT with other extrastriate areas and visual thalamus. The phenomenon of Blindsight continues to fascinate neuroscientists and neurologists interested in consciousness ever since the initial discovery by Weisskrantz et al. and is of clinical relevance for cortical blindness resulting from stroke or atrophy. This new study provides an important link between earlier monkey work that relied on experimental lesions and the human Blindsight work with neurological cases of this study's first author (Bridge). The paper is well written and presents a number of intriguing observations that are suitably contextualized and discussed by the authors in terms of relevant studies in both humans and monkeys. The new research is important, timely and fills a critical gap in the literature. However, the reviewers have expressed important concerns that need to be addressed before the manuscript can be accepted for publication.

Essential revisions:

1) The lesion extent needs to more clearly be described. What exactly do the authors mean by “almost complete loss of V1”? The lesion description is too vague: “A structural MRI identified bilateral hemianopia with almost complete loss of primary visual cortex (V1) (Figure 1, top)”. From Figure 1 it is clear that the monkey has enlarged ventricles that appear to affect predominantly the ventral occipital lobe. But for the reader it is impossible to determine which areas (in addition to V1?) are affected, how they are affected (thinning of gray matter?). There is some calcarine clearly present in the top left (sagittal) image of animal S in Figure 1, and more importantly, some of the occipital pole surface cortex does seem to be present in the both sagittal and horizontal images of Monkey S; in the myelin-weighted horizontal image of Figure 2, that occipital pole tissue even seems to look much like that of the controls. From this MR image it appears that opercular / parafoveal V1 (my estimate is up to ~8 degrees eccentricities) seems intact and the lesion might primarily affect higher eccentricities. Without additional information (e.g. histology), describing V1 as "missing" (or implying that it's completely nonfunctional) seems premature. This leaves the possibility that any visual guided behavior (or the face patch activation in Figure 10) is carried out by some parts of V1 that are currently not clear due to insufficient anatomical description and insensitive fMRI methodology.

2) The authors need to clearly demonstrate that there is no or only very spatially confined BOLD activation in V1. It seems that the current protocol is insensitive to reveal reliable V1 BOLD activation even in normal animals. The variability in activation across the controls and the many areas of puzzling activation in some of them detract from the believability of small areas of activation in Monkey S (or any of the monkeys) as being something meaningful and specific. For example, cortical activation illustrated for the flickering checkerboard (Figure 3) is not only (maybe) "less consistent" across the controls than activation in LGN, it seems to encompass numerous nonvisual areas in Monkey M902. Face stimuli (Figure 10) seem to activate a wide swath of nonvisual cortex laterally in the same monkey, and parts of the caudate, insula, etc., in M901. It would be good to know a bit more about why there seem to be these spurious (?) activations so that we can better trust that the activations identified as specific face patches or portions of MT are indeed "signal". In addition, I infer from the "Visual Stimuli" section that only some of the controls showed visual responses in the LGN at all to the flickering checkerboard. In general, the paper would benefit from a section in the Discussion explicitly devoted to limitations on interpretation given variability among the controls and discussing possible artifacts or other sources of activations that may not be meaningful. The authors might provide a more substantial series of sections (ideally in all three planes) illustrating the lack of BOLD activation in these potential spared portions of V1.

3) In the absence of a clearer demonstration of Blindsight, the title needs to be revised. "Blindsight" seems inappropriate on two counts: not only does it appear possible that a substantial portion of V1 is spared (see above), but also the term "blindsight” has a contentious history as applied to monkeys, as the authors probably know. In terms of the behavior, Blindsight, even in monkeys, has been taken to mean a dissociation between essentially reflexive behavior and behavior that reflects some judgment of the absence of the stimulus. E.g., responding to a peripheral target when cued vs. not responding when there is no cue or categorizing a stimulus in the same way as a blank trial. Assuming that even if monkey S could be trained on tasks that would provide such a dissociation, those types of training often take months even with animals with (partial) visual lesions and a (partial) fovea or pseudofovea. However, the authors could include something like "Implications for blindsight" as the terminal part of their title (i.e., it's not the word blindsight that's the problem, it's what can and can't be inferred about it from the study that's at issue). A more accurate overall title would be something like "Preserved extrastriate visual network in a monkey with naturally occurring substantial damage to V1."

4) A more complete behavioral description of S. In the present form of the manuscript, there is no description of the psychophysical testing procedure in the Materials and methods section and the reader has no chance to form his own impression and has to rely on the authors description. Suggestions include a video of S (although there is also acknowledgement that this might be problematic) and determining if the animal can fixate. At minimum, a more detailed description of the behavioral tasks that were already used is needed and additional behavioral documentation is encouraged.

[Editors' note: further revisions were requested prior to acceptance, as described below.]

Thank you for resubmitting your work entitled "Preserved extrastriate visual network in a monkey with substantial, naturally occurring damage to primary visual cortex" for further consideration at *eLife*. Your revised article has been favorably evaluated by Joshua Gold (Senior Editor), a Reviewing Editor, and two reviewers.

The manuscript has been improved but there are some remaining issues that need to be addressed before acceptance, as outlined below:

1) Abstract: the authors should more accurately describe their functional activation pattern. In light of the existing literature with acquired lesions in monkeys and humans, the lack of functional activation of area MT (and presumably other retinotopic areas) should be spelled out in parallel with the presence of such activation of the face patches in the temporal lobe. In light of this activation pattern in monkey S, the authors should be cautious about their conclusion of 'an intact network of visual cortical areas even without V1', which appears to draw primarily on the functional connectivity analysis.

2) Face patch activation: Please specify where the patches in this study are located in reference to the pattern reported by Tsao/Freiwald and others. Many colleagues would find this information very useful. In the Materials and methods section the authors mention neutral vs threatening expressions – are there any activation differences associated with this (Hoffman et al., 2007)?

3) Figure 2: Thank you for providing this post mortem scan. While this helps to appreciate the reduced gray matter thickness of V1, I am wondering about the variability across V1. In particular the lateral region (foveal representation) appears thicker, but this may just be an impression.

4) Figure 1—figure supplement 1: It is not clear how the authors derive the eccentricities for the expected visual field loss.

5) Introduction paragraph four: LGN also receives SC input (Harting, 1991; Stepniewska, Qi and Kaas, 1999)

6) Subsection “Case history of monkey S” paragraph two: Did this training on oneplace in the lab involve head fixation and eye movement control?

7) Discussion paragraph three: I believe stronger wording could be justified: our study (Schmid et al., 2010) did not only suggest this, but actually tested and demonstrated the effect of LGN inactivation on behaviour and cortical activation. Although you could argue that the inactivation might also include pulvinar due to its proximity to LGN, and as we did not present results from pulvinar testing.

8) Subsection “Limitations of analysing visual activation under general anaesthesia”: but the use of anaesthesia does not explain the residual activation in the temporal lobe

9) Abstract: I completely concur with including a mention of blindsight here, but in the new version the term seems a bit shoehorned into the second sentence, interrupting the expository flow of the Abstract. The authors might consider removing the "including blindsight" in that sentence, and instead conclude the Abstract with a statement about the results "having implications for residual visual capacities even if they are not conscious, as in the blindsight phenomenon" (or something to that effect – nothing too strong, but still including and even highlighting the blindsight connection a bit more, if desired).

10) Figure 2: Which piece of the image are the white arrows pointing at exactly? The figure legend says the stripe of Gennari, but it is very hard to see anything within the gray matter strip corresponding to cortex. Or are the arrows meant to demarcate the extent (borders) of V1 as inferred by the stria of Gennari? That would not seem to entirely make sense in terms of the arrow placement.

---

## [Author Response]

Essential revisions:1) The lesion extent needs to more clearly be described. What exactly do the authors mean by “almost complete loss of V1”? The lesion description is too vague: “A structural MRI identified bilateral hemianopia with almost complete loss of primary visual cortex (V1) (Figure 1, top)”. From Figure 1 it is clear that the monkey has enlarged ventricles that appear to affect predominantly the ventral occipital lobe. But for the reader it is impossible to determine which areas (in addition to V1?) are affected, how they are affected (thinning of gray matter?). There is some calcarine clearly present in the top left (sagittal) image of animal S in Figure 1, and more importantly, some of the occipital pole surface cortex does seem to be present in the both sagittal and horizontal images of Monkey S; in the myelin-weighted horizontal image of Figure 2, that occipital pole tissue even seems to look much like that of the controls. From this MR image it appears that opercular / parafoveal V1 (my estimate is up to ~8 degrees eccentricities) seems intact and the lesion might primarily affect higher eccentricities. Without additional information (e.g. histology), describing V1 as "missing" (or implying that it's completely nonfunctional) seems premature. This leaves the possibility that any visual guided behavior (or the face patch activation in Figure 10) is carried out by some parts of V1 that are currently not clear due to insufficient anatomical description and insensitive fMRI methodology.

We have now conducted a set of high resolution post mortem scans on the intact brain situated in the skull at 7T. It is unclear at this stage how useful the histology would be, as the remaining tissue at the occipital pole is very thin and fragile – therefore we focussed on the post mortem high resolution (0.27x0.27x0.27mm3) scans at 7T. Evidence of the fragility is the partial collapse of the cortical ribbon shown in the post mortem image. In the paper, we now include additional results and further images of V1 at high resolution, a more detailed description of the extent of lesion, measurements of cortical thickness and the visibility of the stria of gennari in the putative remains of V1 shown as new Figure 2 (and see Legend and Results subsection “Cortical structure: V1 appears much thinner in monkey S but area V5/MT shows similar pattern

of dense myelination to control monkeys”). We also added a control hemisphere scanned with the same protocol for comparison.

We also make the anatomical MRI source data available through the Oxford Wellcome Imaging Centre (WIN) GitHub https://git.fmrib.ox.ac.uk/hb/BrainWithoutV1, so that readers can form their own views and critically check our data directly.

2) The authors need to clearly demonstrate that there is no or only very spatially confined BOLD activation in V1. It seems that the current protocol is insensitive to reveal reliable V1 BOLD activation even in normal animals. The variability in activation across the controls and the many areas of puzzling activation in some of them detract from the believability of small areas of activation in Monkey S (or any of the monkeys) as being something meaningful and specific. For example, cortical activation illustrated for the flickering checkerboard (Figure 3) is not only (maybe) "less consistent" across the controls than activation in LGN, it seems to encompass numerous nonvisual areas in Monkey M902. Face stimuli (Figure 10) seem to activate a wide swath of nonvisual cortex laterally in the same monkey, and parts of the caudate, insula, etc., in M901. It would be good to know a bit more about why there seem to be these spurious (?) activations so that we can better trust that the activations identified as specific face patches or portions of MT are indeed "signal". In addition, I infer from the "Visual Stimuli" section that only some of the controls showed visual responses in the LGN at all to the flickering checkerboard. In general, the paper would benefit from a section in the Discussion explicitly devoted to limitations on interpretation given variability among the controls and discussing possible artifacts or other sources of activations that may not be meaningful. The authors might provide a more substantial series of sections (ideally in all three planes) illustrating the lack of BOLD activation in these potential spared portions of V1.

We agree that it is important to document the patterns of activity (or the lack thereof) more fully, so that the reader can check our reported results for themselves. We have now added a complete series of 1mm-spaced sections through the whole brain for animal S for both the checkerboard and the motion stimulus (see Figure 4—figure supplement 2, Figure 5—figure supplement 1). These document the patterns of activity in V1, V5/MT and the rest of the brain comprehensively across the whole brain. Since all regions of interest are shown, we did not feel it was necessary to provide the other planes. We also make the functional imaging source data available online for monkey S and the controls through the Oxford Wellcome Imaging Centre GitHub https://git.fmrib.ox.ac.uk/hb/BrainWithoutV1.

We also amended what is now Figure 6, to show the quantitative activation in V1 for monkey S and across all of the controls. We provide in the text the ratio of V1/LGN activation for all monkeys included in this study (Results subsection “LGN shows significant activation to visual stimulation in monkey S”, paragraph four).

These results underpin that in monkey S, there is no activation of V1, which contrast with patchy activation in dorsal STS and the face patches in ventral STS.

We have also amended the Discussion with a new section for a more explicit debate about the limitations of our BOLD results given the variability among the controls and discussing possible artefactual activations we might see (Discussion subsection “Limitations of analysing visual activation under general anaesthesia”).

3) In the absence of a clearer demonstration of Blindsight, the title needs to be revised. "Blindsight" seems inappropriate on two counts: not only does it appear possible that a substantial portion of V1 is spared (see above), but also the term "blindsight” has a contentious history as applied to monkeys, as the authors probably know. In terms of the behavior, Blindsight, even in monkeys, has been taken to mean a dissociation between essentially reflexive behavior and behavior that reflects some judgment of the absence of the stimulus. E.g., responding to a peripheral target when cued vs. not responding when there is no cue or categorizing a stimulus in the same way as a blank trial. Assuming that even if monkey S could be trained on tasks that would provide such a dissociation, those types of training often take months even with animals with (partial) visual lesions and a (partial) fovea or pseudofovea. However, the authors could include something like "Implications for blindsight" as the terminal part of their title (i.e., it's not the word blindsight that's the problem, it's what can and can't be inferred about it from the study that's at issue). A more accurate overall title would be something like "Preserved extrastriate visual network in a monkey with naturally occurring substantial damage to V1."

We have changed the title to ‘Preserved extrastriate visual network in a monkey with substantial, naturally occurring damage to primary visual cortex’. We mention ‘blindsight’ into the Abstract and the keywords, so that the article can be more easily found by the relevant communities.

4) A more complete behavioral description of S. In the present form of the manuscript, there is no description of the psychophysical testing procedure in the Materials and methods section and the reader has no chance to form his own impression and has to rely on the authors description. Suggestions include a video of S (although there is also acknowledgement that this might be problematic) and determining if the animal can fixate. At minimum, a more detailed description of the behavioral tasks that were already used is needed and additional behavioral documentation is encouraged.

Thank you for pointing out this omission – we have now added a complete description of the training regime and attempted task training with animal S in the Materials and methods section (new section ‘Behavioural training and tasks’). Where available, we also added quantitative measures of task performance in the Results (see subsection “Case history of monkey S”). We also added further details of a clinical, behavioural observation by a neurologist carried out at the time. We regret to say that we have no video material and the animal is not alive anymore.

[Editors' note: further revisions were requested prior to acceptance, as described below.]

The manuscript has been improved but there are some remaining issues that need to be addressed before acceptance, as outlined below:1) Abstract: the authors should more accurately describe their functional activation pattern. In light of the existing literature with acquired lesions in monkeys and humans, the lack of functional activation of area MT (and presumably other retinotopic areas) should be spelled out in parallel with the presence of such activation of the face patches in the temporal lobe. In light of this activation pattern in monkey S, the authors should be cautious about their conclusion of 'an intact network of visual cortical areas even without V1', which appears to draw primarily on the functional connectivity analysis.

We have revised the Abstract – including the concluding sentence – in the light of the comments above. We now have an explicit statement of the sparsity of visual cortical activation except for the face patches. We would prefer to be cautious about specifically highlighting the absence of activation in a specific area, especially since we deal here with functional activations under general anaesthesia. In the Abstract, the network result only appears now in its specific analysis context. We would like to point out that extrastriate visual cortical areas, particularly V5/MT appear also structurally intact, see myelin results.

We did our very best include all the requested elements of the results in the Abstract’s 150 words. See also Comment 9 for further requested changes to the Abstract. See new Abstract:

“Lesions of primary visual cortex (V1) lead to loss of conscious visual perception with significant impact on human patients. Understanding the neural consequences of such damage may aid the development of rehabilitation methods. In this rare case of a Rhesus macaque (monkey S), likely born without V1, the animal’s in-group behaviour was unremarkable, but visual task training was impaired. With multi-modal magnetic resonance imaging, visual structures outside of the lesion appeared normal. Visual stimulation under anaesthesia with checkerboards activated lateral geniculate nucleus of monkey S, while full-field moving dots activated pulvinar. Visual cortical activation was sparse but included face patches. Consistently across lesion and control monkeys, functional connectivity analysis revealed an intact network of bilateral dorsal visual areas temporally correlated with V5/MT activation, even without V1. Despite robust subcortical responses to visual stimulation, we found little evidence for strengthened subcortical input to V5/MT supporting residual visual function or blindsight-like phenomena.”

2) Face patch activation: Please specify where the patches in this study are located in reference to the pattern reported by Tsao/Freiwald and others. Many colleagues would find this information very useful. In the Materials and methods section the authors mention neutral vs threatening expressions – are there any activation differences associated with this (Hoffman et al., 2007)?

We have set our results more specifically into the context of the Tsao/Freiwald pattern (naming AF and MF as the relevant patches). We would prefer to be cautious in interpreting these results, as we mentioned in the manuscript that we have only two controls available and the specific activations are small.

We did not find evidence for specific activations for threatening vs neutral faces in the STS in line with the results in awake Rhesus macaque reported by Hoffman et al., 2007 and Hadj-Bouziane et al., 2008. We have amended the Results accordingly. See last paragraph of Results:

“When we compared the BOLD response to all face stimuli compared to a mid-gray background (similar to the localizer used by Liu et al., 2015, we found clearly defined clusters of activation for the two most commonly identified temporal lobe face patches in the Rhesus macaque: the anterior (likely AF, Tsao et al., 2008, and middle (likely MF, Tsao et al., 2008) face patch located in the fundus of ventral STS (Figure 11). One control showed only the anterior face patch. While the locations of these areas are as previously described in macaques (Tsao et al., 2003, Pinsk et al., 2005, Tsao et al., 2008, Bell et al., 2009), we did not check the contrast against scrambled faces. We did not observe any specific differences in activation between threatening and neutral facial expressions in the STS, consistent with previous studies (Hoffman et al., 2007, Hadj-Bouziane et al., 2008).”

There might be a small activation near the amygdala, but we had two experts in MRI face responses check the scans and they thought that these specific responses in this small data set obtained under general anaesthesia might be too weak for a meaningful comparison on this subtle difference.

We also provide the full functional scan data through a weblink, so readers can inspect the original data in full:

https://web.gin.g-node.org/hbridge_oxford/brainwithoutv1

3) Figure 2: Thank you for providing this post mortem scan. While this helps to appreciate the reduced gray matter thickness of V1, I am wondering about the variability across V1. In particular the lateral region (foveal representation) appears thicker, but this may just be an impression.

On further inspection, this is a consequence of the angle of the cut through the cortical ribbon at this point. We have amended the figure legend to make readers aware of this. See Figure legends: Figure 2:

“Figure 2: A post mortem T2*-weighted structural scan at high field (7T) revealed the stripe of Gennari in V1 (white arrows for example sites) of both monkey S and a control monkey (M131), although the position relative to the cortical surface within the cortical ribbon appeared to differ. The V1 gray matter in monkey S was reduced in thickness to 0.9-1.3 mm around the enlarged ventricles (1.4-1.8 mm in the control). The appearance of increased thickness more laterally in the sulcus is due to the angle of the slice through cortex at this point. By contrast, the gray matter at the location for V5/MT appeared intact with a typical thickness of 1.9-2.0 mm. Note that the lesion is partially light gray in appearance as well partially black. As the brain was still inside the skull during the scan, we speculate that this might be a result of slow incursion of ‘fluorinert’, in which the skull was immersed for this scan.”

We also provide the full scan data through a weblink, so readers can inspect the original data in full:

https://web.gin.g-node.org/hbridge_oxford/brainwithoutv1

4) Figure 1—figure supplement S1: It is not clear how the authors derive the eccentricities for the expected visual field loss.

We estimated visual eccentricities based on the cortical folding pattern and slice position using the neurophysiological map of retinotopy obtained by van Essen et al., 1984 (see their Figure 11). We have clarified this in the figure legend. See Figure legends: Figure 1–figure supplement 1:

“Figure 1—figure supplement 1: Based on established neurophysiological maps of V1 retinotopy in relation to the cortical folding pattern and approximate slice position (their Figure 11, Van Essen et al., 1984), we estimated the visual field loss of monkey S. Visual map is shown on post mortem T2*-weighted images – deformations of the occipital cortical surface were post mortem. Given the extent of the loss of white matter directly underneath V1, we expect bilateral visual field loss, for the central 10° as well as for more peripheral parts, potentially around 15°-20° eccentricity.”

5) Introduction paragraph four: LGN also receives SC input (Harting, 1991; Stepniewska, Qi andKaas, 1999)

Thank you for the references, which we have added:

“In addition to its direct retinal input, the LGN also receives projections from the superior colliculus (SC) (Harting et al., 1991, Stepniewska et al., 1999). The pulvinar nucleus receives input from the superior colliculus (SC), although probably not in the most appropriate subdivision for projecting to V5/MT, or surviving early projections directly from the retina (Stepniewska et al., 1999, Warner et al., 2010).”

6) Subsection “Case history of monkey S” paragraph two: Did this training on oneplace in the lab involve head fixation and eye movement control?

Monkey S did not have a head implant. Whether in the transport box or upright in the primate chair, head fixation and eye movement tracking could not be implemented. We have now made this more explicit in the text:

“At this phase of the training, monkey S was transferred to a primate chair (seated, but without head clamp or eye movement control) and to a different reward schedule (juice instead of pellets), and ‘oneplace’ was attempted again.”

7) Discussion paragraph three: I believe stronger wording could be justified: our study (Schmid et al., 2010) did not only suggest this, but actually tested and demonstrated the effect of LGN inactivation on behaviour and cortical activation. Although you could argue that the inactivation might also include pulvinar due to its proximity to LGN, and as we did not present results from pulvinar testing.

We have changed the wording as requested (p11, lines 353-354). The sentence now reads:

“Combined lesion and silencing studies in macaques reinforce the requirement for an intact LGN to support remaining visual function after V1 lesions (Schmid et al., 2010).”

8) Subsection “Limitations of analysing visual activation under general anaesthesia”: but the use of anaesthesia does not explain the residual activation in the temporal lobe

This is correct and we now point this out more explicitly in the Discussion:

“The level of BOLD activation is affected by general anaesthesia parameters such as type and depth (Hutchison et al., 2014, Vincent et al., 2007), although previous results do not explain the potential differences in dorsal and temporal visual activations we see.”

We also explain in the same section that we cannot exclude that the level of anaesthesia might vary at different times during the scan.

9) Abstract: I completely concur with including a mention of blindsight here, but in the new version the term seems a bit shoehorned into the second sentence, interrupting the expository flow of the Abstract. The authors might consider removing the "including blindsight" in that sentence, and instead conclude the Abstract with a statement about the results "having implications for residual visual capacities even if they are not conscious, as in the blindsight phenomenon" (or something to that effect – nothing too strong, but still including and even highlighting the blindsight connection a bit more, if desired).

As suggested, we have removed ‘including blindsight’ from the 2nd sentence and worked it in at the end, albeit a bit shorter than suggested because of the 150 words limitation. We agree that this has improved the flow of the Abstract. (See Abstract – under point 1 above – for changes).

10) Figure 2: Which piece of the image are the white arrows pointing at exactly? The figure legend says the stripe of Gennari, but it is very hard to see anything within the gray matter strip corresponding to cortex. Or are the arrows meant to demarcate the extent (borders) of V1 as inferred by the stria of Gennari? That would not seem to entirely make sense in terms of the arrow placement.

We are sorry that this was unclear. The arrow placement was at a selection of example points where we identified the stripe of Gennari in the image. We clarify this now in figure legend 2 (see full figure legend under point 3 above).

We checked the original figure and the embedded figure in our Word document both on screen and as printouts. We agree that especially the Word embedded version was too small and of too low quality to identify the stripe of Gennari clearly. We have now adjusted the overall brightness of the displayed images and provide them as high resolution image at full page size (see new Figure 2). We have also repositioned one arrow to a clearer site.

Again, since these data are available on

https://web.gin.g-node.org/hbridge_oxford/brainwithoutv1

and the reader is free to explore the location.